# CDT: Cascading Decision Trees for Explainable Reinforcement Learning

## Abstract

Deep Reinforcement Learning (DRL) has recently achieved significant advances in various domains. However, explaining the policy of RL agents still remains an open problem due to several factors, one being the complexity of explaining neural networks decisions. Recently, a group of works have used decision-tree-based models to learn explainable policies. Soft decision trees (SDTs) and discretized differentiable decision trees (DDTs) have been demonstrated to achieve both good performance and share the benefit of having explainable policies. In this work, we further improve the results for tree-based explainable RL in both performance and explainability. Our proposal, Cascading Decision Trees (CDTs) apply representation learning on the decision path to allow richer expressivity. Empirical results show that in both situations, where CDTs are used as policy function approximators or as imitation learners to explain black-box policies, CDTs can achieve better performances with more succinct and explainable models than SDTs. As a second contribution our study reveals limitations of explaining black-box policies via imitation learning with tree-based explainable models, due to its inherent instability.

## 1 Introduction

Explainable Artificial Intelligence (XAI), especially Explainable Reinforcement Learning (XRL) (Puiutta and Veith, 2020) is attracting more attention recently. How to interpret the action choices in reinforcement learning (RL) policies remains a critical challenge, especially as the gradually increasing trend of applying RL in various domains involving transparency and safety (Cheng et al., 2019; Junges et al., 2016). Currently, many state-of-the-art DRL agents use neural networks (NNs) as their function approximators. While NNs are considered stronger function approximators (for better performances), RL agents built on top of them are generally lack of interpretability (Lipton, 2018). Indeed, interpreting the behavior of NNs themselves remains an open problem in the field (Montavon et al., 2018; Albawi et al., 2017).

In contrast, traditional DTs (with hard decision boundaries) are usually regarded as models with readable interpretations for humans, since humans can interpret the decision making process by visualizing the decision path. However, DTs may suffer from weak expressivity and therefore low accuracy. An early approach to reduce the hardness of DT was the soft/fuzzy DT (shorten as SDT) proposed by Suárez and Lutsko (1999). Recently, differentiable SDTs (Frosst and Hinton, 2017) have shown both improved interpretability and better function approximation, which lie in the middle of traditional DTs and neural networks.

People have adopted differentiable DTs for interpreting RL policies in two slightly different settings: an imitation learning setting (Coppens et al., 2019; Liu et al., 2018), in which imitators with interpretable models are learned from RL agents with black-box models, or a full RL setting (Silva et al., 2019), where the policy is directly represented as an interpretable model, *e.g.*, DT. However, the DTs in these methods only conduct partitions in raw feature spaces without representation learning that could lead to complicated combinations of partitions, possibly hindering both model interpretability and scalability. Even worse, some methods have axis-aligned partitions (univariate decision nodes) (Wu et al., 2017; Silva et al., 2019) with much lower model expressivity.

In this paper, we propose Cascading Decision Trees (CDTs) striking a balance between model interpretability and accuracy, this is, having an adequate representation learning based on interpretable models (*e.g.* linear models). Our experiments show that CDTs share the benefits of having a significantly smaller number of parameters (and a more compact tree structure) and better performance than related works. The experiments are conducted on RL tasks, in either imitation-learning or RL settings. We also demonstrate that the imitation-learning approach is less reliable for interpreting the

RL policies with DTs, since the imitating DTs may be prominently different in several runs, which also leads to divergent feature importances and tree structures.

## 2 RELATED WORKS

A series of works were developed in the past two decades along the direction of differentiable DTs (Irsoy et al., 2012; Laptev and Buhmann, 2014). Recently, Frosst and Hinton (2017) proposed to distill a SDT from a neural network. Their approach was only tested on MNIST digit classification tasks. Wu et al. (2017) further proposed the tree regularization technique to favor the models with decision boundaries closer to compact DTs for achieving interpretability. To further boost the prediction accuracy of tree-based models, two main extensions based on single SDT were proposed: (1) ensemble of trees, or (2) unification of NNs and DTs.

An ensemble of decision trees is a common technique used for increasing accuracy or robustness of prediction, which can be incorporated in SDTs (Rota Bulo and Kontschieder, 2014; Kontschieder et al., 2015; Kumar et al., 2016), giving rise to neural decision forests. Since more than one tree needs to be considered during the inference process, this might yield complications in the interpretability. A common solution is to transform the decision forests into a single tree (Sagi and Rokach, 2020).

As for the unification of NNs and DTs, Laptev and Buhmann (2014) propose convolutional decision trees for feature learning from images. Adaptive Neural Trees (ANTs) (Tanno et al., 2018) incorporate representation learning in decision nodes of a differentiable tree with nonlinear transformations like convolutional neural networks (CNNs). The nonlinear transformations of an ANT, not only in routing functions on its decision nodes but also in feature spaces, guarantee the prediction performances in classification tasks on the one hand, but also hinder the potential of interpretability of such methods on the other hand. Wan et al. (2020) propose the neural-backed decision tree (NBDT) which transfers the final fully connected layer of a NN into a DT with induced hierarchies for the ease of interpretation, but shares the convolutional backbones with normal deep NNs, yielding the state-of-the-art performances on CIFAR10 and ImageNet classification tasks.

However, these advanced methods either employ multiple trees with multiplicative numbers of model parameters, or heavily incorporate deep learning models like CNNs in the DTs. Their interpretability is severely hindered due to their model complexity.

To interpret an RL agent, Coppens et al. (2019) propose distilling the RL policy into a differentiable DT by imitating a pre-trained policy. Similarly, Liu et al. (2018) apply an imitation learning framework but to the $Q$ value function of the RL agent. They also propose Linear Model U-trees (LMUTs) which allow linear models in leaf nodes. Silva et al. (2019) propose to apply differentiable DTs directly as function approximators for either $Q$ function or the policy in RL. They apply a discretization process and a rule list tree structure to simplify the trees for improving interpretability. The VIPER method proposed by Bastani et al. (2018) also distills policy as NNs into a DT policy with theoretically verifiable capability, but for imitation learning settings and nonparametric DTs only.

Our proposed CDT is distinguished from other main categories of methods with differentiable DTs for XRL in the following ways: (i) Compared with SDT (Frosst and Hinton, 2017), partitions in CDT not only happen in original input space, but also in transformed spaces by leveraging intermediate features. This is well documented in recent works (Kontschieder et al., 2015; Xiao, 2017; Tanno et al., 2018) to improve model capacity, and it can be further extended into hierarchical representation learning with advanced feature learning modules like CNN (Tanno et al., 2018). (ii) Compared with work by Coppens et al. (2019), space partitions are not limited to axis-aligned ones (which hinders the expressivity of trees with certain depths), but achieved with linear models of features as the routing functions. Moreover, the adopted linear models are not a restriction (but as an example) and other interpretable transformations are also allowed in our CDT method. (iii) Compared with ANTs (Tanno et al., 2018), our CDT method unifies the decision making process based on different intermediate features with a single decision making tree, which follows the low-rank decomposition of a large matrix with linear models. It thus greatly improves the model simplicity for achieving interpretability. About model simplicity and interpretability in DTs, see our motivating example in Appendix A.

## 3 METHOD

### 3.1 SOFT DECISION TREE (SDT)

A SDT is a differentiable DT with a probabilistic decision boundary at each node. Considering we have a DT of depth $D$, each node in the SDT can be represented as a weight vector (with the bias as

an additional dimension) $\boldsymbol{w}_i^j$, where $i$ and $j$ indicate the index of the layer and the index of the node in that layer respectively, as shown in Fig. 1. The corresponding node is represented as $n_u$, where $u = 2^{i-1} + j$ uniquely indices the node.

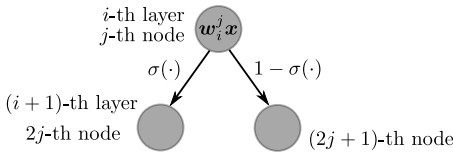

Figure 1: A SDT node. $\sigma(\cdot)$ is the sigmoid function with function values on decision nodes as input.

The decision path for a single instance can be represented as set of nodes $\mathcal{P} \subset \mathcal{N}$, where $\mathcal{N}$ is the set for all nodes on the tree. We have $\mathcal{P} = \arg\max_{\{u\}} \prod_{i=1}^D p_{i-1\to i}^{\lfloor j/2 \rfloor \to j}$, where $p_{i-1\to i}^{\lfloor j/2 \rfloor \to j}$ is the probability of going from node $n_{2^{i-2}+\lfloor j/2 \rfloor}$ to $n_{2^{i-1}+j}$. The $\{u\}$ indicates that the $\arg\max$ is taken over a set of nodes rather than a single one. Note that $p_{i-1\to i}^{\lfloor j/2 \rfloor \to j}$ will always be 1 for a hard DT (Safavian and Landgrebe, 1991). Therefore the path probability to a specific node $n_u$ is: $P^u = \prod_{i'=1}^{j'} p_{i'-1\to i'}^{\lfloor j'/2 \rfloor \to j'}, u' \in \mathcal{P}$. In the following, we name all DTs using probabilistic decision path as SDT-based methods, shorten as SDT.

Silva et al. (2019) propose to discretize the learned differentiable SDTs into univariate DTs for improving interpretability. Specifically, for a decision node with a $(k+1)$-dimensional vector $\boldsymbol{w}$ (the first dimension $w_1$ is the bias term), the discretization process (i) selects the index of largest weight dimension as $k^* = \arg\max_k w_k$ and (ii) divides $w_1$ by $w_{k^*}$, to construct a univariate hard DT. Without further description, the default discretization process in our experiments for both SDTs and CDTs also follows this manner. The SDTs are therefore the same as DDTs in Silva et al. (2019).

### 3.2 CASCADING DECISION TREE (CDT)

**Methods.** We propose CDT[1] as an extension based on SDT, allowing it to have the capability of representation learning as well as decision making in transformed spaces. In a simple CDT architecture as shown on the left of Fig. 2, a feature learning tree $\mathcal{F}$ is cascaded with a decision making tree $\mathcal{D}$. In tree $\mathcal{F}$, each decision node is a simple function of raw feature vector $\boldsymbol{x}$ given learnable parameters $\boldsymbol{w}$: $\phi(\boldsymbol{x}; \boldsymbol{w})$, while each leaf of it is a feature representation function: $\boldsymbol{f} = f(\boldsymbol{x}; \tilde{\boldsymbol{w}})$ parameterized by $\tilde{\boldsymbol{w}}$. In tree $\mathcal{D}$, each decision node is a simple function of learned features $\boldsymbol{f}$ rather than raw features $\boldsymbol{x}$ given learnable parameters $\boldsymbol{w}'$: $\psi(\boldsymbol{f}; \boldsymbol{w}')$. The output distribution of $\mathcal{D}$ is another parameterized function $p(\cdot; \tilde{\boldsymbol{w}}')$ independent of either $\boldsymbol{x}$ or $\boldsymbol{f}$. For simplicity and interpretability, all functions $\phi, f$ and $\psi$ are linear functions in our examples, but they are free to be extended with other interpretable models.

Specifically, we provide detailed mathematical relationships based on linear functions as follows. For an environment with input state vector $\boldsymbol{x}$ and output discrete action dimension $O$, suppose that our CDT has intermediate features of dimension $K$ (not the number of leaf nodes on $\mathcal{F}$, but for each leaf node), we have the probability of going to the left/right path on the $u$-th node on $\mathcal{F}$:

$$p_u^{\text{Go Left}} = \sigma(\boldsymbol{w}_k \cdot \boldsymbol{x}), \quad p_u^{\text{Go Right}} = 1 - p_u^{\text{Go Left}}, \tag{1}$$

which is the same as in SDTs. Then we have the linear feature representation function for each leaf node on $\mathcal{F}$, which transforms the basis of the representation space with:

$$f_k = \tilde{\boldsymbol{w}}_k \cdot \boldsymbol{x}, k = 0, 1, ..., K-1 \tag{2}$$

which gives the $K$-dimensional intermediate feature vector $\boldsymbol{f}$ for each possible path. Due to the symmetry in all internal layers within a tree, all internal nodes satisfy the formulas in Eq. (1)(2). In tree $\mathcal{D}$, it is also a SDT but with raw input $\boldsymbol{x}$ replaced by learned representations $\boldsymbol{f}$ for each node $u'$ in $\mathcal{D}$:

$$p_{u'}^{\text{Go Left}} = \sigma(\tilde{\boldsymbol{w}}_k \cdot \boldsymbol{f}), \quad p_{u'}^{\text{Go Right}} = 1 - p_{u'}^{\text{Go Left}}, \tag{3}$$

---

[1]A motivation of CDT method is the duplicative structures in the heuristic solution of *LunarLander-v2*, as discussed in Appendix B

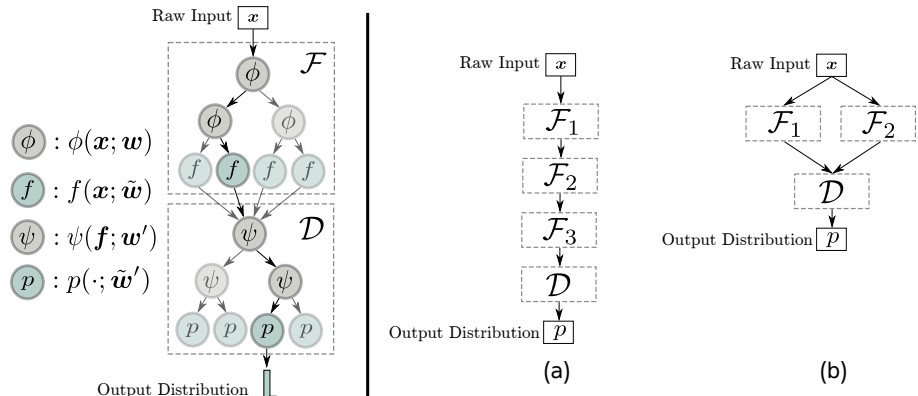

Figure 2: CDT methods. Left: a simple CDT architecture, consisting of a feature learning tree $\mathcal{F}$ and a decision making tree $\mathcal{D}$; Right: two possible types of hierarchical CDT architectures, where (a) is an example architecture with hierarchical representation learning using three cascading $\mathcal{F}$ before one $\mathcal{D}$, and (b) is an example architecture with two $\mathcal{F}$ in parallel, potentially with different dimensions of $x$ as inputs.

Finally, the output distribution is feature-independent, which gives the probability mass values across output dimension $O$ for each leaf of $\mathcal{D}$ as:

$$p_{k'} = \frac{\exp(\tilde{\boldsymbol{w}}')}{\sum_{k'=0}^{O-1} \exp(\tilde{\boldsymbol{w}}'_{k'})}, k' = 0, 1, ..., O-1 \tag{4}$$

Suppose we have a CDT of depth $N_1$ for $\mathcal{F}$ and depth $N_2$ for $\mathcal{D}$, the probability of going from root of either $\mathcal{F}$ or $\mathcal{D}$ to $u$-th leaf node on each sub-tree both satisfies previous derivation in SDTs: $P^u = \prod_{i'=1}^{j'} p_{i'-1 \to i'}^{\lfloor j'/2 \rfloor \to j'}, u' \in \mathcal{P}$, where $\mathcal{P}$ is the set of nodes on path. Therefore the overall path probability of starting from the root of $\mathcal{F}$ to $u_1$-th leaf node of $\mathcal{F}$ and then $u_2$-th leaf node of $\mathcal{D}$ is:

$$P = P^{u_1} P^{u_2} \tag{5}$$

Each leaf of the feature learning tree represents one possible assignment for intermediate feature values, while they share the subsequent decision making tree. During the inference process, we simply take the leaf on $\mathcal{F}$ or $\mathcal{D}$ with the largest probability to assign values for intermediate features (in $\mathcal{F}$) or derive output probability (in $\mathcal{D}$), which may sacrifice little accuracy but increase interpretability. The detailed architecture of CDT with relationships among variables is plotted in figures in Appendix C.

**Model Simplicity**

We analyze the simplicity of CDT compared with SDT in terms of the numbers of learnable parameters in the model. The reason for doing this is that in order to increase the interpretability, we need to simplify the tree structure or reduce the number of parameters including weights and bias in the tree.

We can analyze the model simplicity of CDT against a normal SDT with linear functions in a matrix decomposition perspective. Suppose we need a total of $M$ multivariate decision nodes in the $R$-dimensional raw input space $\mathbb{X}$ to successfully partition the space for high-performance prediction, which can be written as a matrix $\mathbf{W}_{M \times R}^x$. CDT tries to achieve the same partitions through learning a transformation matrix $\mathbf{T}_{K \times R} : \mathbb{X} \to \mathbb{F}$ for all leaf nodes in $\mathcal{F}$ and a partition matrix $\mathbf{W}_{M \times K}^f$ for all internal nodes in $\mathcal{D}$ in the $K$-dimensional feature space $\mathbb{F}$, such that:

$$\mathbf{W}^x \boldsymbol{x} = \mathbf{W}^f \boldsymbol{f} = \mathbf{W}^f \mathbf{T} \boldsymbol{x} \tag{6}$$

$$\Rightarrow \mathbf{W}^x = \mathbf{W}^f \mathbf{T} \tag{7}$$

Therefore the number of model parameters to be learned with CDT is reduced by $M \times R - (M \times K + K \times R)$ compared against a standard SDT of the same total depth, and it is a positive value as long as $K < \frac{M \times R}{M + R}$, while keeping the model expressivity.

A detailed quantitative analysis of model parameters for CDT and SDT is provided in Appendix D.

**Hierarchical CDT**

From above, a simple CDT architecture as in Fig. 2 with a single feature learning model $\mathcal{F}$ and single decision making model $\mathcal{D}$ can achieve intermediate feature learning with a significant reduction in model complexity compared with traditional SDT. However, sometimes the intermediate features learned with $\mathcal{F}$ may be unsatisfying for capturing complex structures in advanced tasks, therefore we further extend the simple CDT architecture into more hierarchical ones. As shown on the right side in Fig. 2, two potential types of hierarchical CDT are displayed: (a) a hierarchical feature abstraction module with three feature learning models $\{\mathcal{F}_1, \mathcal{F}_2, \mathcal{F}_3\}$ in a cascading manner before inputting to the decision module $\mathcal{D}$; (b) a parallel feature extraction module with two feature learning models $\{\mathcal{F}_1, \mathcal{F}_2\}$ before concatenating all learned features into $\mathcal{D}$.

One needs to bear in mind that whenever the model structures are complicating, the interpretability of the model decreases due to the loss of simplicity. Therefore we did not apply the hierarchical CDTs in our experiments for maintaining interpretability. However, the hierarchical structure is one of the most preferred ways to keep simplicity as much as possible if trying to increase the model capacity and prediction accuracy, so it can be applied when necessary.

# 4 EXPERIMENTS

We compare CDT and SDT on two settings for interpreting RL agents: (1) the imitation learning setting, whereas the RL agent with a black-box model (*e.g.* neural network) to interpret first generates a state-action dataset for imitators to learn from, and the interpretation is derived on the imitators; (2) the full RL setting, whereas the RL agent is directly trained with the policy represented with interpretable models like CDTs or SDTs, such that the interpretation can be derived by directly spying into those models. The environments are *CartPole-v1*, *LunarLander-v2* and *MountainCar-v0* in OpenAI Gym (Brockman et al., 2016). The depth of CDT is represented as "$d_1 + d_2$" in the following sections, where $d_1$ is the depth of feature learning tree $\mathcal{F}$ and $d_2$ is the depth of decision making tree $\mathcal{D}$. Each setting is trained for five runs in imitation learning and three runs in RL.

Both the fidelity and stability of mimic models reflect the reliability of them as interpretable models. Fidelity is the accuracy of the mimic model, *w.r.t.* the original model. It is an estimation of similarity between the mimic model and the original one in terms of prediction results. However, fidelity is not sufficient for reliable interpretations. An unstable family of mimic models will lead to inconsistent explanations of original black-box models. The stability of the mimic model is a deeper excavation into the model itself and comparisons among several runs. Previous research (Bastani et al., 2017) has investigated the fidelity and stability of decision trees as mimic models, where the stability is estimated with the fraction of equivalent nodes in different random decision trees trained under the same settings. In our experiments, the stability analysis is conducted via comparing tree weights of different instances in imitation learning settings.

## 4.1 IMITATION LEARNING

**Performance.** The datasets for imitation learning are generated with heuristic agents for environments *CartPole-v1* and *LunarLander-v2*, containing 10000 episodes of state-action data for each environments. See Appendix E for other training details. The results are provided in Table 1 and 2.

| Tree Type | Depth | Discretized | Accuracy (%) | Episode Reward | # of Parameters |
|---|---|---|---|---|---|
| SDT | 2 | ✗ | 94.1±0.01 | 500.0±0.0 | 23 |
| | | ✓ | 49.7±0.02 | 39.9±7.6 | 14 |
| | 3 | ✗ | 94.5±0.1 | 500.0±0.0 | 51 |
| | | ✓ | 50.0±0.01 | 42.5±7.3 | 30 |
| | 4 | ✗ | 94.3±0.3 | 500.0±0.0 | 107 |
| | | ✓ | 50.1±0.1 | 40.4±7.8 | 62 |
| CDT (ours) | 1+2 | ✗ | 95.4±1.1 | 500.0±0.0 | 38 |
| | | $\mathcal{F}$ only | 94.4±0.8 | 500.0±0.0 | 35 |
| | | $\mathcal{D}$ only | 84.1±2.8 | 500.0±0.0 | 35 |
| | | $\mathcal{F} + \mathcal{D}$ | 83.8±2.6 | 497.8±8.4 | 32 |
| | 2+1 | ✗ | 95.6±0.1 | 500.0±0.0 | 54 |
| | | $\mathcal{F}$ only | 92.7±0.4 | 500.0±0.0 | 45 |
| | | $\mathcal{D}$ only | 88.4±1.3 | 500.0±0.0 | 53 |
| | | $\mathcal{F} + \mathcal{D}$ | 89.0±0.4 | 500.0±0.0 | 44 |
| | 2+2 | ✗ | 96.6±0.9 | 500.0±0.0 | 64 |
| | | $\mathcal{F}$ only | 91.6±1.3 | 500.0±0.0 | 55 |
| | | $\mathcal{D}$ only | 82.9±3.7 | 494.8±19.8 | 61 |
| | | $\mathcal{F} + \mathcal{D}$ | 81.9±1.8 | 488.8±31.4 | 52 |

Table 1: Comparison of CDT and SDT with imitation-learning settings on *CartPole-v1*.

CDTs perform consistently better than SDTs before and after discretization process in terms of prediction accuracy, with different depths of the tree. Additionally, for providing a similarly accurate model, CDT method always has a much smaller number of parameters compared with SDT, which improves its interpretability as shown in later sections. However, although better than SDTs, CDTs also suffer from degradation in performance after discretization, which could lead to unstable and unexpected models. We claim that this is a general drawback for tree-based methods with soft decision boundaries in XRL with imitation-learning settings, which is further studied in the following.

| Tree Type | Depth | Discretized | Accuracy (%) | Episode Reward | # of Parameters |
|---|---|---|---|---|---|
| SDT | 4 | ✗ | 85.4±0.4 | 58.2±246.1 | 199 |
| | | ✓ | 54.8±10.1 | -237.1±121.9 | 94 |
| | 5 | ✗ | 87.6±0.5 | 191.3±143.8 | 407 |
| | | ✓ | 51.6±4.5 | -93.7±102.9 | 190 |
| | 6 | ✗ | 88.7±1.3 | 193.4±161.4 | 823 |
| | | ✓ | 60.2±3.9 | -172.4±122.0 | 382 |
| | 7 | ✗ | 88.9±0.5 | 194.2±138.8 | 1655 |
| | | ✓ | 62.7±2.8 | -233.4±62.4 | 766 |
| CDT (ours) | 2+2 | ✗ | 88.2±1.6 | 107.4±190.7 | 116 |
| | | $\mathcal{F}$ only | 78.0±2.4 | -126.9±237.0 | 95 |
| | | $\mathcal{D}$ only | 68.3±10.3 | -301.6±136.8 | 113 |
| | | $\mathcal{F} + \mathcal{D}$ | 64.4±12.1 | -229.7±256.0 | 92 |
| | 2+3 | ✗ | 88.3±1.7 | 168.5±169.0 | 144 |
| | | $\mathcal{F}$ only | 70.2±2.3 | -9.7±159.2 | 123 |
| | | $\mathcal{D}$ only | 40.7±11.9 | -106.3±187.7 | 137 |
| | | $\mathcal{F} + \mathcal{D}$ | 35.9±1.5 | -130.2±135.9 | 116 |
| | 3+2 | ✗ | 90.4±1.7 | 199.5±123.7 | 216 |
| | | $\mathcal{F}$ only | 72.2±8.3 | -14.2±175.6 | 167 |
| | | $\mathcal{D}$ only | 78.1±2.5 | 150.8±148.1 | 209 |
| | | $\mathcal{F} + \mathcal{D}$ | 64.6±4.7 | 7.1±173.6 | 160 |
| | 3+3 | ✗ | 90.4±1.2 | 173.0±124.5 | 244 |
| | | $\mathcal{F}$ only | 72.0±1.2 | -55.3±178.6 | 195 |
| | | $\mathcal{D}$ only | 58.7±8.6 | -91.5±97.0 | 237 |
| | | $\mathcal{F} + \mathcal{D}$ | 46.8±5.6 | -210.5±121.9 | 188 |

Table 2: Comparison of CDT and SDT with imitation-learning settings on *LunarLander-v2*.

**Stability.** To investigate the stability of imitation learners for interpreting the original agents, we measure the normalized weight vectors from different imitation-learning trees. For SDTs, the weight vectors are the linear weights on inner nodes, while for CDTs $\{\tilde{w}, \tilde{w}'\}$ are considered. Through the experiments, we would like to show how unstable the imitators $\{\mathbf{L}\}$ are. We have a tree agent $\mathbf{X} \in \{\mathbf{L}', \mathbf{H}, \mathbf{R}\}$, where $\mathbf{L}'$ is another imitator tree agent trained under the same setting, $\mathbf{R}$ is a random tree agent, and $\mathbf{H}$ is a heuristic tree agent (used for generating the training dataset). The distances of tree weights between two agents $\mathbf{L}, \mathbf{X}$ are calculated with the following formula:

$$D(\mathbf{L}, \mathbf{X}) = \frac{1}{2N} \sum_{n=1}^{N} \min_{m=1,2,...,M} ||\boldsymbol{l}_m - \boldsymbol{x}_n||_1 + \frac{1}{2M} \sum_{m=1}^{M} \min_{n=1,2,...,N} ||\boldsymbol{x}_m - \boldsymbol{l}_n||_1 \quad (8)$$

while $\overline{D(\mathbf{L}, \mathbf{X})}$ are averaged over all possible $\mathbf{L}$s and $\mathbf{X}$s with the same setting. Since we have the heuristic agent for *LunarLander-v2* environment and we transform the heuristic agent into a multivariate DT agent, we get the decision boundaries of the tree on all its nodes. So we also compare the differences of decision boundaries in heuristic tree agent $\mathbf{H}$ and those of the learned tree agent $\mathbf{L}$. But we do not have the official heuristic agent for *CartPole-v1* in the form of a decision tree. For the decision making trees in CDTs, we transform the weights back into the input feature space to make a fair comparison with SDT and the heuristic tree agent. The results are displayed in Table 3, all trees use intermediate features of dimension 2 for both environments. In terms of stability, CDTs generally perform similarly as SDTs and even better on *CartPole-v1* environment.

We further evaluate the feature importance with at least two different methods on SDTs to demonstrate the instability of imitation learning settings for XRL, see Appendix F.1. We also display all trees (CDTs and SDTs) for both environments in Appendix F.3. Significant differences can be found in different runs for the same tree structure with the same training setting, which testifies the unstable and unrepeatable nature by interpreting imitators instead of the original agents.

**Conclusion.** We claim that the current imitation-learning setting with tree-based models is not suitable for interpreting the original RL agent, with the following evidence derived from our experiments: (i) The discretization process usually degrades the performance (prediction accuracy) of the agent significantly, especially for SDTs. Although CDTs alleviate the problem to a certain extent, the performance degradation is still not negligible, therefore the imitators are not expected to be alternatives for interpreting the original agents; (ii) With the stability analysis in our experiments, we find that different imitators will display different tree structures even if they follow the same training setting on the same dataset, which leads to significantly different decision paths and local feature importance assignments.

| Tree Type | Env | Depth | $\overline{D(\mathbf{L}, \mathbf{L}')}$ | $\overline{D(\mathbf{L}, \mathbf{R})}$ | $\overline{D(\mathbf{L}, \mathbf{H})}$ |
|---|---|---|---|---|---|
| SDT | CartPole-v1 | 3 | 0.21 | 0.90±0.10 | - |
| | LunarLander-v2 | 4 | 0.50 | 0.92±0.05 | 0.84 |
| CDT (ours) | CartPole-v1 | 1+2 | 0.07 | 1.05±0.15 | - |
| | | 2+2 | 0.19 | 1.03±0.10 | - |
| | LunarLander-v2 | 2+2 | 0.63 | 1.01±0.10 | 0.98 |
| | | 3+3 | 0.53 | 0.83±0.06 | 0.86 |

Table 3: Tree Stability Analysis. $\overline{D(\mathbf{L}, \mathbf{L}')}$, $\overline{D(\mathbf{L}, \mathbf{R})}$ and $\overline{D(\mathbf{L}, \mathbf{H})}$ are average values of distance between an imitator $\mathbf{L}$ and another imitator $\mathbf{L}'$, or a random agent $\mathbf{R}$, or a heuristic agent $\mathbf{H}$ with metric $D$. CDTs are generally more stable, but still with large variances over different imitators.

## 4.2 REINFORCEMENT LEARNING

**Performance.** We evaluate the learning performances of different DTs and NNs as policy function approximators in RL, as shown in Fig. 3. Every setting is trained for three runs. We use Proximal Policy Optimization (Schulman et al., 2017) algorithm in our experiments. The multilayer perceptron (MLP) model is a two-layer NN with 128 hidden units. The SDT has a depth of 3 for *CartPole-v1* and 4 for *LunarLander-v2*. The CDT has depths of 2 and 2 for feature learning tree and decision making tree respectively on *CartPole-v1*, while with depths of 3 and 3 for *LunarLander-v2*. Therefore for each environment, the SDTs and CDTs have a similar number of model parameters, while MLP model has at least 6 times more parameters. Detailed training settings are provided in Appendix G. From Fig. 3, we can see that CDTs can at least outperform SDTs as policy function approximators for RL in terms of both sampling efficiency and final performance, although may not learn as fast as general MLPs with a significantly larger number of parameters. For *MountainCar-v0* environment, the learning performances are less stable due to the sparse reward signals and large variances in exploration. However, with CDT for policy function approximation, there are still near-optimal agents after training with or without state normalization, as displayed in Appendix H.

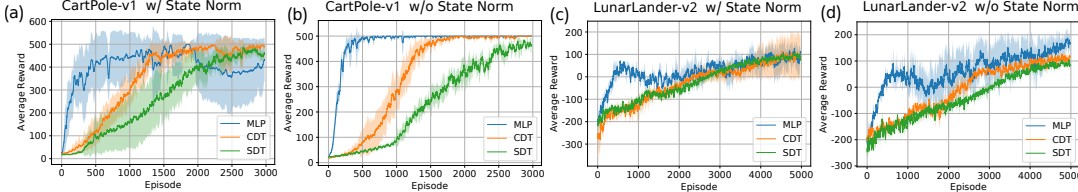

Figure 3: Comparison of SDTs and CDTs on *CartPole-v1* and *LunarLander-v2* in terms of average rewards in RL setting: (a)(c) use normalized input states while (b)(d) use unnormalized ones.

**Tree Depth.** The depths of DTs are also investigated for both SDT and CDT, because deeper trees tend to have more model parameters and therefore lay more stress on the accuracy rather than interpretability. Fig. 4 shows the learning curves of SDTs and CDTs in RL with different tree depths for the two environments, using normalized states as input, while the comparison with unnormalized states is in Appendix H with similar results. From the comparisons, we can see that generally deeper trees can learn faster with even better final performances for both CDTs and SDTs, but CDTs are less sensitive to tree depth than SDTs.

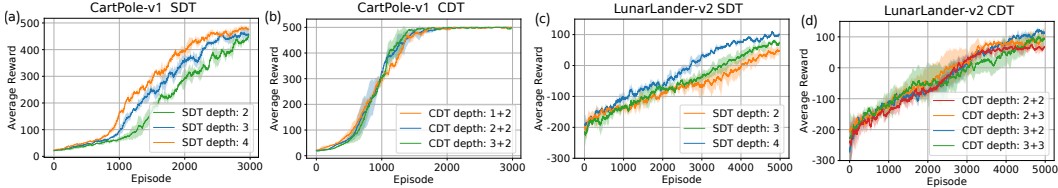

Figure 4: Comparison of SDTs and CDTs with different depths (state normalized). (a) and (b) are trained on *CartPole-v1*, while (c) and (d) are on *LunarLander-v2*.

**Interpretability.** We display the learned CDTs in RL settings for three environments, compared against some heuristic solutions or SDTs. A heuristic solution[2] for *CartPole-v1* is: if $3\theta + \dot{\theta} > 0$,

---

[2]Provided by Zhiqing Xiao on OpenAI Gym Leaderboard: https://github.com/openai/gym/wiki/Leaderboard

push right; otherwise, push left. As shown in Fig. 5, in our learned CDT of depth 1+2, the weights of two-dimensional intermediate features ($f[0]$ and $f[1]$) are much larger on the last two dimensions of observation than the first two, therefore we can approximately ignore the first two dimensions due to their low importance in decision making process. So we get similar intermediate features for two cases in two dimensions, which are approximately $w_1 x[2] + w_2 x[3] \rightarrow w\theta + \dot{\theta}$ after normalization ($w > 0$). Based on the decision making tree in learned CDT, it gives a close solution as the heuristic one, yielding if $w\theta + \dot{\theta} < 0$ push left otherwise push right. The original CDT before discretization and a SDT for comparison are provided in Appendix I.

For *MountainCar-v0*, due to the complexity in the landscape as shown in Fig. 6, interpreting the learned model is even harder. However, through CDT, we can see that the agent learns intermediate features as combinations of car position and velocity, potentially being an estimated future position or previous position, and makes action decisions based on that. The original CDT before discretization has depth 2+2 with one-dimensional intermediate features, and its structure is shown in Appendix I.

Due to the page limitation, the learned CDT as an example for *LunarLander-v2* is provided in Appendix I, which also captures some important feature combinations like the angle with angular speed and X-Y coordinate relationships for decision making.

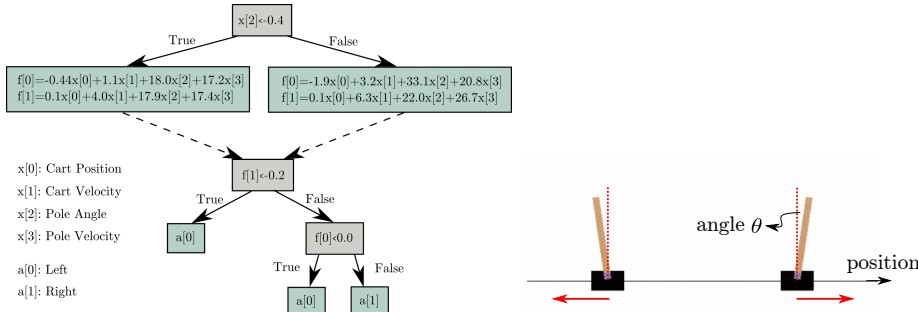

Figure 5: Left: learned CDT (after discretization). Right: game scene of *CartPole-v1*.

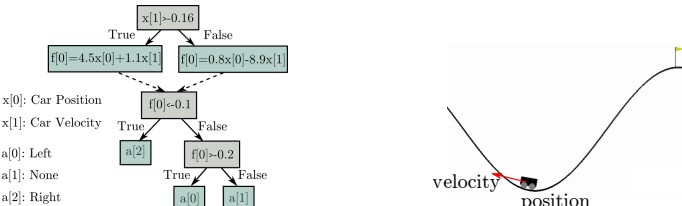

Figure 6: Left: learned CDT (after discretization). Right: game scene of *MountainCar-v0*.

## 5 CONCLUSION

In this work, we have proposed a new architecture of differentiable DT, the Cascading Decision Tree (CDT). A simple CDT cascades a feature learning DT and a decision making DT into a single model. From our experiments, we show that compared with traditional differentiable DTs (*i.e.*, DDTs or SDTs) CDTs have better function approximation in both imitation learning and full RL settings with a significantly reduced number of model parameters while better preserving the tree prediction accuracy after discretization. We also qualitatively and quantitively corroborate that the SDT-based methods with imitation learning setting may not be proper for achieving interpretable RL agents due to instability among different imitators in their tree structures, even when having similar performances. Finally, we contrast the interpretability of learned DTs in RL settings, especially for the intermediate features. Our analysis supports that CDTs lend themselves to be further extended to hierarchical architectures with more interpretable modules, due to its ts richer expressivity allowed via representation learning. More work needs to be done to fully realize the potential of our method, which involves the investigation of hierarchical CDT settings and well-regularized intermediate features for further interpretability. Additionally, since the present experiments are demonstrated with linear transformations in the feature space, non-linear transformations are expected to be leveraged for tasks with higher complexity or continuous action space while preserving interpretability.

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

## A   SIMPLICITY AND INTERPRETABILITY: A MOTIVATING EXAMPLE

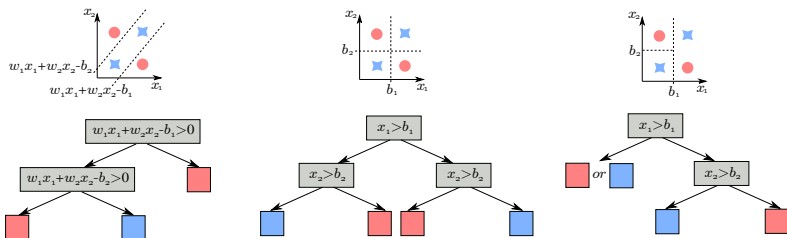

Figure 7: Comparison of three different tree structures on a simple classification problem. From left to right: (1) multivariate DT; (2) univariate DT; (3) differentiable rule lists.

People have proposed a variety of desiderata for interpretability (Lipton, 2018), including trust, causality, transferability, informativeness, etc. Here we summarize the answers in general into two aspects: (1) interpretable meta-variables that can be directly understood; (2) model simplicity. Understandable variables with simple model structures comprise most of the models interpreted by humans either by physical and mathematical principles or human intuitions, which is also in accordance with the Occam's razor principle.

For model simplicity, a simple model in most cases is more interpretable than a complicated one. Different metrics can be applied to measure the model complexity (Murray, 2007; Molnar et al., 2019), like the number of model parameters, model capacity, computational complexity, non-linearity, etc. There are ways to reduce the model complexity: model projection from a large space into a small sub-space, merging the replicates in the model, etc. Feature importance (Schwab and Karlen, 2019) (*e.g.*, through estimating model sensitivity to changes of inputs) is one type of methods for projecting a complicated model into a scalar space across feature dimensions. The proposed method CDT in this paper is a way to improve model simplicity by merging the replicates through representation learning.

For a binary classification problem, three different tree structures and their decision boundaries are compared in Fig. 7: (1) multivariate DT; (2) univariate DT; (3) differentiable rule lists (Silva et al., 2019). We need to define the simplicity of DT for achieving interpretability and choose which type of tree is the one we prefer.

For the first two structures, we may not be able to draw conclusions for their simplicity since it seems one has a simpler tree structure but more complex decision boundaries while the other one is the opposite. We will have another example to clarify it. But we can draw a conclusion for the second

and the third ones since the structure of differentiable rule lists is simpler, as it has an asymmetric structure and the left nodes are always leaves. However, the problem of differentiable rule lists is also obvious, that it sacrifices the model capacity and therefore hurts the accuracy. For the first left node in the example, it can only choose either one of the two labels without distinctiveness, which is clearly not correct.

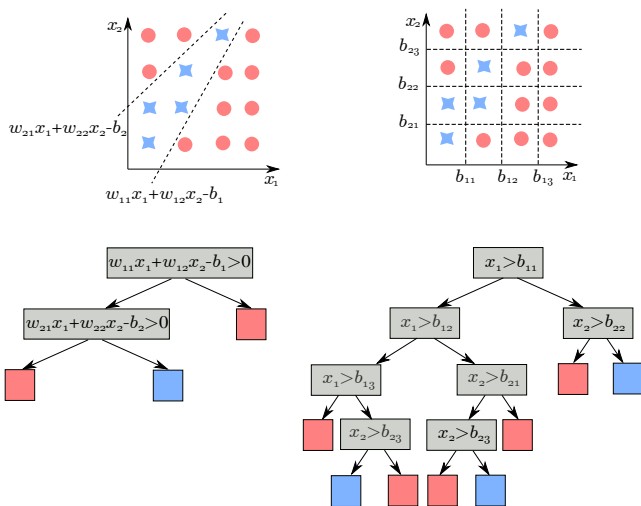

Figure 8: Comparison of two different tree structures on a complicated classification problem.

To clarify the problem of choosing between the first two structures, a more complicated example is provided in Fig. 8. It shows the comparison of a multivariate DT and a univariate DT for a binary classification task. Apparently, the multivariate DT is simpler than the univariate one in its structure. The conclusion is that for complex cases, the multivariate tree structure has greater potentials of achieving necessary space partitioning with simpler model structures.

## B  DUPLICATIVE STRUCTURE IN HEURISTIC AGENTS

As shown in Fig. 9, we found that the heuristic solution[3] for *LunarLander-v2* contains the duplicative structure after being transformed into a decision tree, and the duplicative structure can be leveraged to simplify the learning models. Specifically, the two green modules (feature learning ones) in the tree are basically assigning different values to two intermediate variables ($ht$ and $at$) under different cases, while the grey module (decision making one) takes the intermediate variables to make action selection. Both the decision making module and the second feature learning module are used repeatedly on different branches on the tree, which forms a duplicative structure. This can help with the simplicity and interpretability of the model, which motivates our idea of CDT methods for XRL.

---

[3]In the code repository of OpenAI Gym: https://github.com/openai/gym/blob/master/gym/envs/box2d/lunar_lander.py

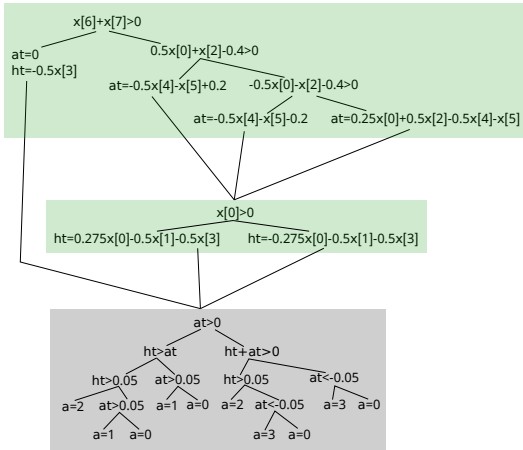

Figure 9: The heuristic decision tree for *LunarLander-v2*. $x$ is an 8-dimensional observation, $a$ is the univariate action given by the agent. $at, ht$ are two intermediate variables, corresponding to the "angle-to-do" and "hover-to-do" in the heuristic solution.

## C    DETAILED SIMPLE CDT ARCHITECTURE

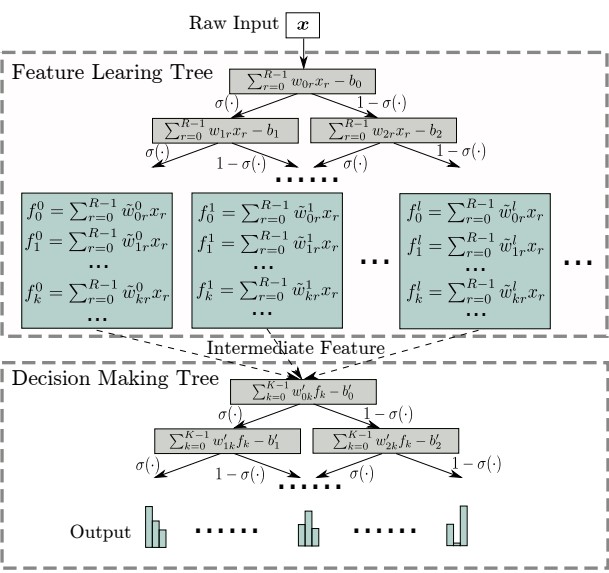

Figure 10: A detailed architecture of simple CDT: the feature learning tree is parameterized by $w$ and $b$, and its leaves are parameterized by $\tilde{w}$ ; the inner nodes of decision making tree are parameterized by $w'$ and $b'$, while the leaves are parameterized by $\tilde{w}'$.

## D    QUANTITATIVE ANALYSIS OF MODEL PARAMETERS

Considering the case where we have a raw feature dimension of inputs as $R$, we choose the intermediate feature dimension to be $K < R$. A CDT with two cascading trees of depth $d_1$ and $d_2$ and a SDT with depth $d$ are compared. Supposing the output dimension is $O$, we can derive the number of parameters in CDT as:

$$N(CDT) = [(R+1)(2^{d_1} - 1) + K \cdot R \cdot 2^{d_1}] + [(K+1)(2^{d_2} - 1) + O \cdot 2^{d_2}] \quad (9)$$

while the number of parameters in SDT is:

$$N(SDT) = (R+1)(2^d - 1) + O \cdot 2^d \quad (10)$$

Considering an example for Eq. 9 and Eq. 10 with SDT being depth of 5 while CDT has $d_1 = 2, d_2 = 3$, raw feature dimension $R = 8$, intermediate feature dimension $K = 4$, and output dimension $O = 4$, we can get $N(CDT) = 222$ and $N(SDT) = 343$. It indicates a reduction of around 35% parameters in this case, which will significantly increase interpretability.

In another example, when $R = 8, K = 4, O = 4$, the numbers of parameters in CDT or SDT models are compared in Fig. 11, assuming $d = d_1 + d_2$ for a total depth of range 2 to 20. The Ratio of numbers of model parameters is derived with: $\frac{N(CDT)}{N(SDT)}$.

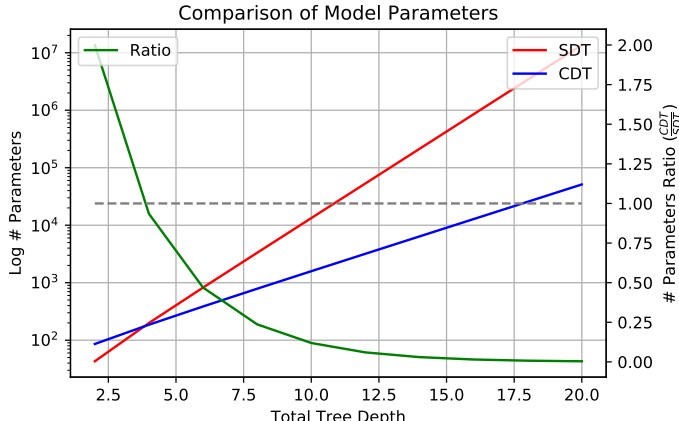

Figure 11: Comparison of numbers of model parameters in CDTs and SDTs. The left vertical axis is the number of model parameters in $\log$-scale. The right vertical axis is the ratio of model parameter numbers. CDT has a decreasing ratio of model parameters against SDT as the total depth of model increases.

## E    HYPERPARAMETERS IN IMITATION LEARNING

| Tree Type | Env | Hyperparameter | Value |
|---|---|---|---|
| Common | CartPole-v1 | learning rate | $1 \times 10^{-3}$ |
| | | batch size | 1280 |
| | | epochs | 80 |
| | LunarLander-v2 | learning rate | $1 \times 10^{-3}$ |
| | | batch size | 1280 |
| | | epochs | 80 |
| SDT | CartPole-v1 | depth | 3 |
| | LunarLander-v2 | depth | 4 |
| CDT | CartPole-v1 | FL depth | 2 |
| | | DM depth | 2 |
| | | # intermediate variables | 2 |
| | LunarLander-v2 | FL depth | 3 |
| | | DM depth | 3 |
| | | # intermediate variables | 2 |

Table 4: Imitation learning hyperparameters. The "Common" hyperparameters are shared for both SDT and CDT.

## F    ADDITIONAL IMITATION LEARNING RESULTS FOR STABILITY ANALYSIS

Both the fidelity and stability of mimic models reflect the reliability of them as interpretable models. Fidelity is the accuracy of the mimic model, *w.r.t.* the original model. It is an estimation of similarity between the mimic model and the original one in terms of prediction results. However, fidelity is not sufficient for reliable interpretations. An unstable family of mimic models will lead to inconsistent explanations of original black-box models. The stability of the mimic model is a deeper excavation

into the model itself and comparisons among several runs. Previous research (Bastani et al., 2017) has investigated the fidelity and stability of decision trees as mimic models, where the stability is estimated with the fraction of equivalent nodes in different random decision trees trained under the same settings. However, in our tests, apart from evaluating the tree weights in different imitators, we also use the feature importance given by different differentiable DT instances with the same architecture and training setting to measure the stability.

### F.1 FEATURE IMPORTANCE ASSIGNMENT ON TREES

For differentiable DT methods (e.g. CDT and SDT), since the decision boundaries within each node are linear combinations of features, we can simply take the weight vector $w_i^j$ as the importance assignment for those features within each node.

After training the tree, a *local explanation* is straightforward to derive with the inference process of a single instance and the decision path on the tree. A *global explanation* can be the average local explanation across instances, *e.g.* in an episode or several episodes under the RL settings. Here we list several ways of assigning importance values for input features with SDT, to derive the feature importance vector $I$ with the same dimension as the decision node vectors $w$ and input feature:

For *local explanation*:

- I. A trivial way of feature importance assignment on SDT would be simply adding up all weight vectors of nodes on the decision path: $I(x) = \sum_{i,j} w_i^j(\boldsymbol{x})$

- II. The second way is a weighted average of the decision vectors, *w.r.t.* the confidence of the decision boundaries for a specific instance. Considering the soft decision boundary on each node, we assume that the more confident the boundary is applied to partition the data point into a specific region within the space, the more reliable we can assign feature importance according to the boundary. The *confidence* of a decision boundary can be positively correlated with the distance from the data point to the boundary, or the probability of the data point falling into one side of the boundary. The latter one is straightforward in our settings. We define the confidence as $p(x) = p_{i-1 \to i}^{\lfloor j/2 \rfloor \to j}(x)$, which is also the probability of choosing node $j$ in $i$-th layer from its parent on instance $x$'s decision path. It indicates how far the data point is from the middle of the soft boundary in a probabilistic view. Therefore the importance value is derived via multiplying the confidence value with each decision node vector: $I(\boldsymbol{x}) = \sum_{i,j} p_{i-1 \to i}^{\lfloor j/2 \rfloor \to j}(\boldsymbol{x}) w_i^j(\boldsymbol{x})$.

  Fig. 12 helps to demonstrate the reason for using the decision confidence (*i.e.*, probability) as a weight for assigning feature importance, which indicates that the probability of belonging to one category is positively correlated with the distance from the instance to the decision boundary. Therefore when there are multiple boundaries for partitioning the space (*e.g.*, two in the figure), we assign the boundaries having a shorter distance to the data point with smaller confidence in determining feature importance, since based on the closer boundaries the data point is much easier to be perturbed into the contrary category and less confident to remain in the original.

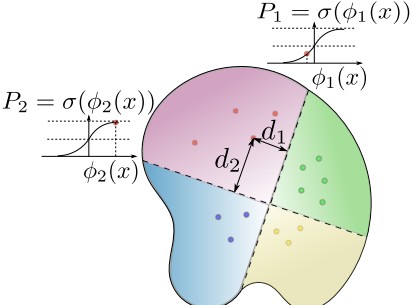

Figure 12: Multiple soft decision boundaries (dashed lines) partition the space. The dots represent input data points, and different colored regions indicate different partitions in the input space. The boundaries closer to the instance are less important in determining the feature importance since they are less distinctive for the instance.

- III. Since the tree we use is differentiable, we can also apply gradient-based methods for feature importance assignment, which is: $\boldsymbol{I}(\boldsymbol{x}) = \frac{\partial y}{\partial \boldsymbol{x}}$, where $y = \text{SDT}(\boldsymbol{x})$.

For *global explanation*:

- We can simply average the feature importance at each time step (*i.e.*, local explanation) to get global feature importance over an episode or across episodes, where the local explanations can be derived in either of the above ways.

### F.2    RESULTS OF FEATURE IMPORTANCE IN IMITATION LEARNING

To testify the stability of applying SDT method with imitation learning from a given agent, we compare the SDT agents of different runs and original agents using certain metrics. The agent to be imitated from is a heuristic decision tree (HDT) agent, and the metric for evaluation is the assigned feature importance across an episode on each feature dimension. As described in the previous section, the feature importance for local explanation can be achieved in three ways, which work for both HDT and SDT here. The environment is *LunarLander-v2* with an 8-dimensional observation in our experiments here.

Considering SDT of different runs may predict different actions, even if they are trained with the same setting and for a considerable time to achieve similarly high accuracies, we conduct comparisons not only for an online decision process during one episode, but also on a pre-collected offline state dataset by the HDT agent. We hope this can alleviate the accumulating differences in trajectories caused by consecutively different actions made by different agents, and give a more fair comparison on the decision process (or feature importance) for the same trajectory.

**Different Tree Depths.** First, the comparison of feature importance (adding up node weights on decision path) for HDT and the learned SDT of different depths in an online decision episode is shown as Fig. 13. All SDT agents are trained for 40 epochs to convergence. The accuracies of three trees are $87.35\%, 95.23\%, 97.50\%$, respectively.

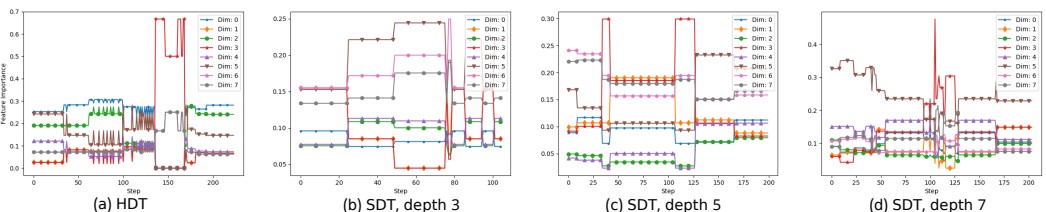

(a) HDT    (b) SDT, depth 3    (c) SDT, depth 5    (d) SDT, depth 7

Figure 13: Comparison of feature importance (local explanation I) for SDT of depth 3, 5, 7 with HDT on an episodic decision making process.

From Fig. 13 we can tell significant differences among SDTs with different depths, as well as in comparing them against the HDT even on the episode with the same random seed, which indicates that the depth of SDT will not only affect the model prediction accuracy but also the decision making process.

**Same Tree with Different Runs.** We compare the feature importance on an offline dataset, containing the states of the HDT agent encounters in one episode. All SDT agents have a depth of 5 and are trained for 80 epochs to convergence. The three agents have testing accuracies of $95.88\%, 97.93\%,$ and $97.79\%$ respectively after training. The feature importance values are evaluated with different approaches as mentioned above (*local explanation* I, II and III) on the same offline episode, as shown in Fig 14. In the results, *local explanation* II and III looks similar, since most decision nodes in the decision path with greatest probability have the probability values close to 1 (*i.e.* close to a hard decision boundary) when going to the child nodes.

From Fig. 14, considerable differences can also be spotted in different runs for local explanations, even though the SDTs have similar prediction accuracies, no matter which metric is applied.

### F.3    TREE STRUCTURES IN IMITATION LEARNING

We display the agents trained with CDTs and SDTs on both *CartPole-v1* and *LunarLander-v2* before and after tree discretization in this section, as in Fig. 15, 16, 17, 18, 20, 21, 22. Each figure contains

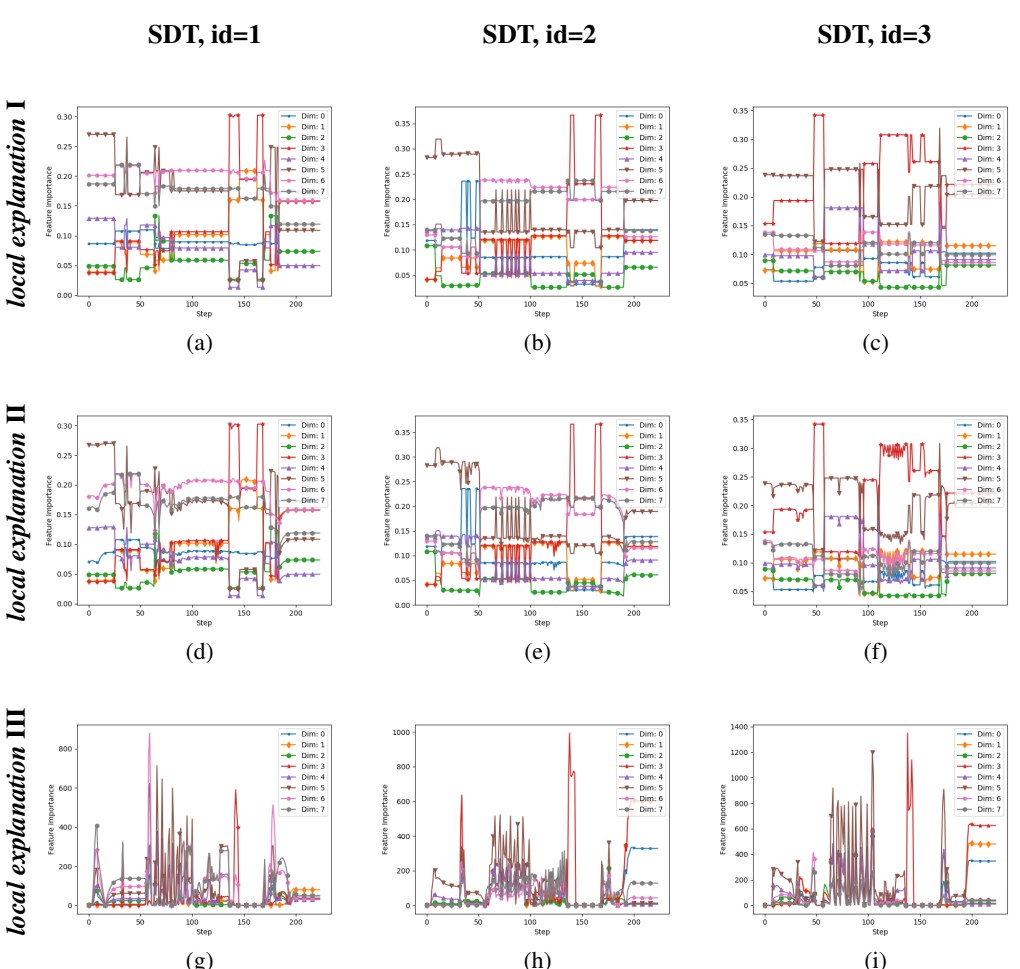

Figure 14: Comparison of feature importance for three SDTs (depth=5, trained under the same setting) with three different local explanations. All runs are conducted on the same offline episode.

trees trained in four runs with the same setting. Each sub-figure contains one learned tree (plus an input example and its output) with an inference path (*i.e.*, the solid lines) for the same input instance. The lines and arrows indicate the connections among tree nodes. The colors of the squares on tree nodes show the values of weight vectors for each node. For feature learning trees in CDTs, the leaf nodes are colored with the feature coefficients. The output leaf nodes of both SDTs and decision making trees in CDTs are colored with the output categorical distributions. Three color bars are displayed on the left side for inputs, tree inner nodes, and output leaves respectively, as demonstrated in Fig. 15. It remains the same for the rest tree plots. The digits on top of each node represent the output action categories.

Among all the learned tree structures, significant differences can be told from weight vectors, as well as intermediate features in CDTs, even if the four trees are under the same training setting. This will lead to considerably different explanations or feature importance assignments on trees.

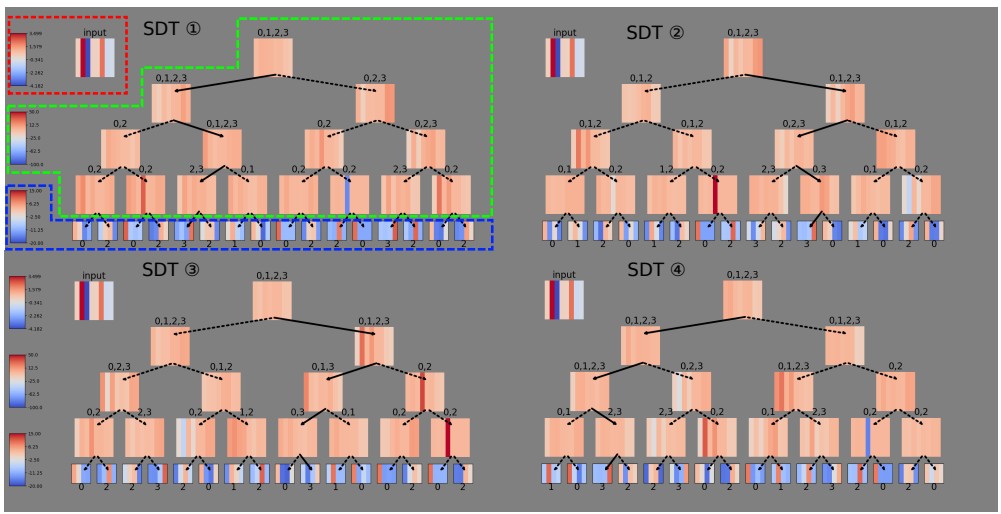

Figure 15: Comparison of four runs with the same setting for SDT (before discretization) imitation learning on *LunarLander-v2*. The dashed lines with different colors on the left top diagram indicate the valid regions for each color bar, which is the default setting for the rest diagrams.

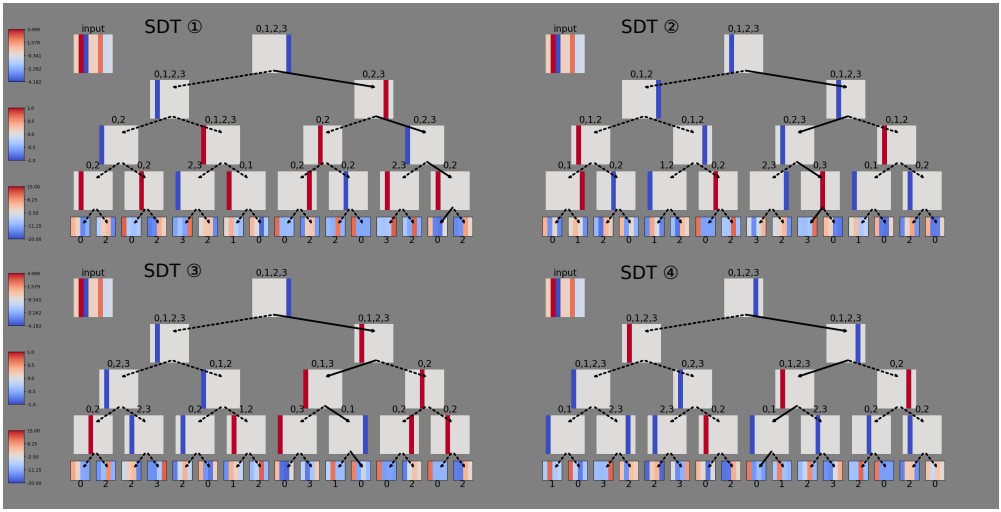

Figure 16: Comparison of four runs with the same setting for SDT (after discretization) imitation learning on *LunarLander-v2*.

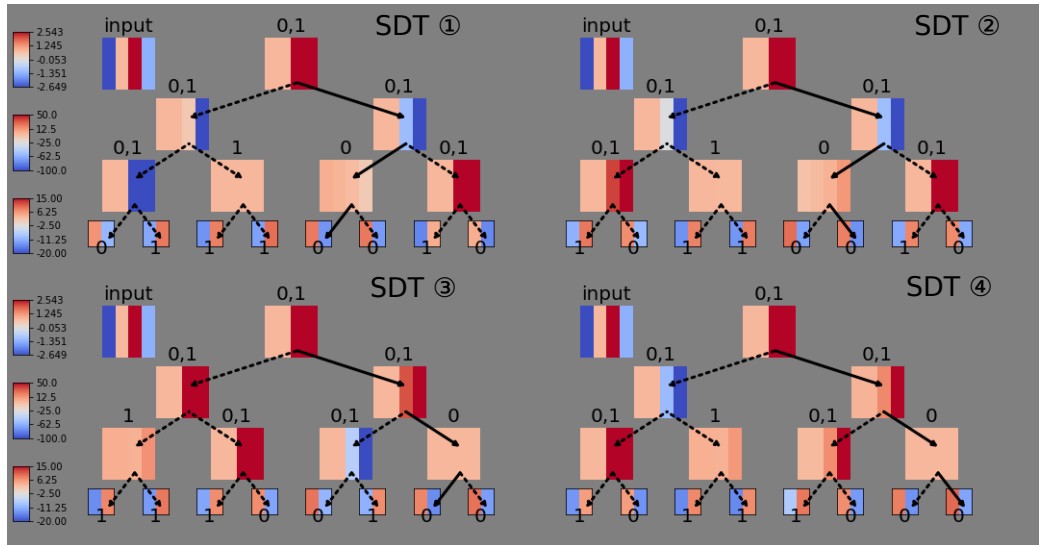

Figure 17: Comparison of four runs with the same setting for SDT (before discretization) imitation learning on *CartPole-v1*.

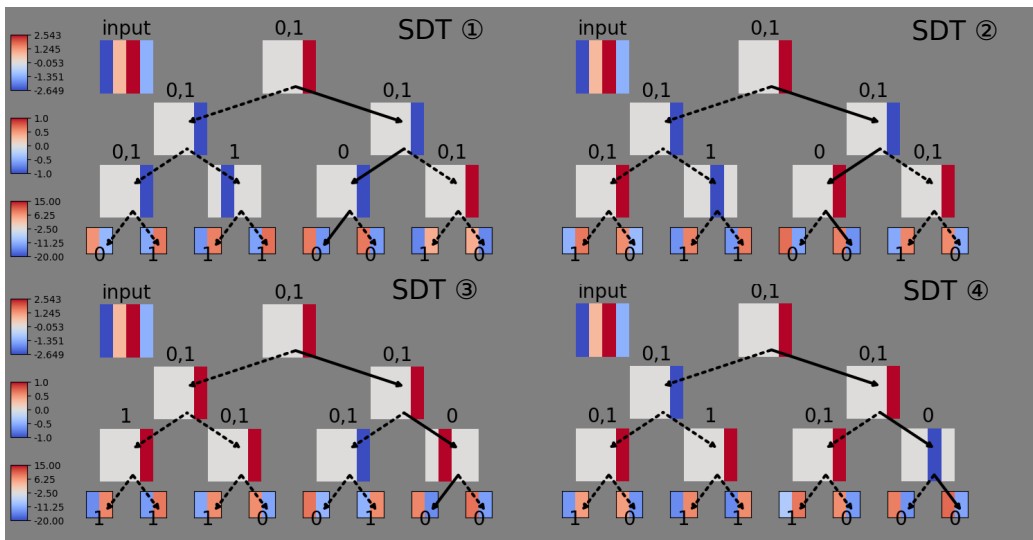

Figure 18: Comparison of four runs with the same setting for SDT (after discretization) imitation learning on *CartPole-v1*.

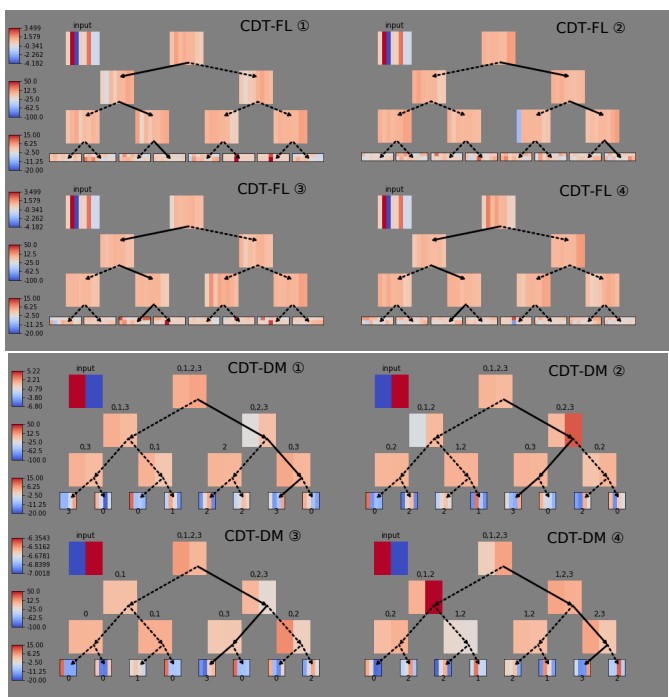

Figure 19: Comparison of four runs with the same setting for CDT (before discretization) imitation learning on *LunarLander-v2*: feature learning trees (top) and decision making trees (bottom).

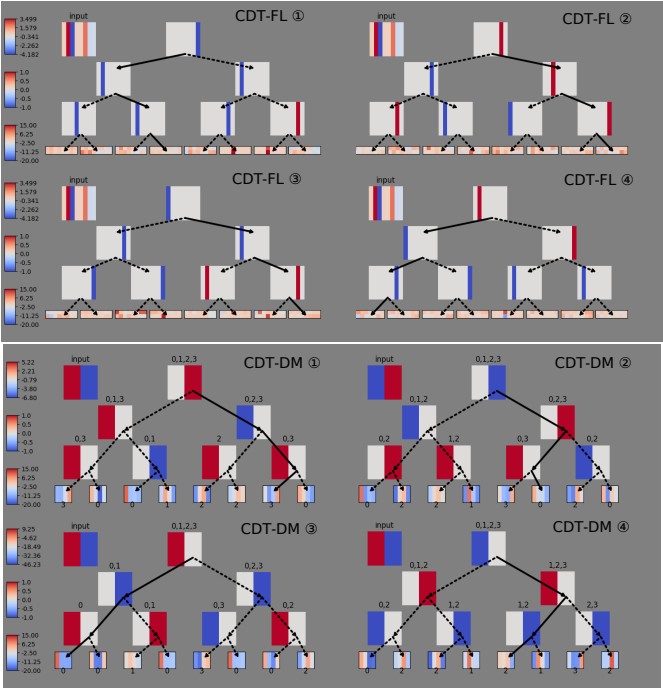

Figure 20: Comparison of four runs with the same setting for CDT (after discretization) imitation learning on *LunarLander-v2*: feature learning trees (top) and decision making trees (bottom).

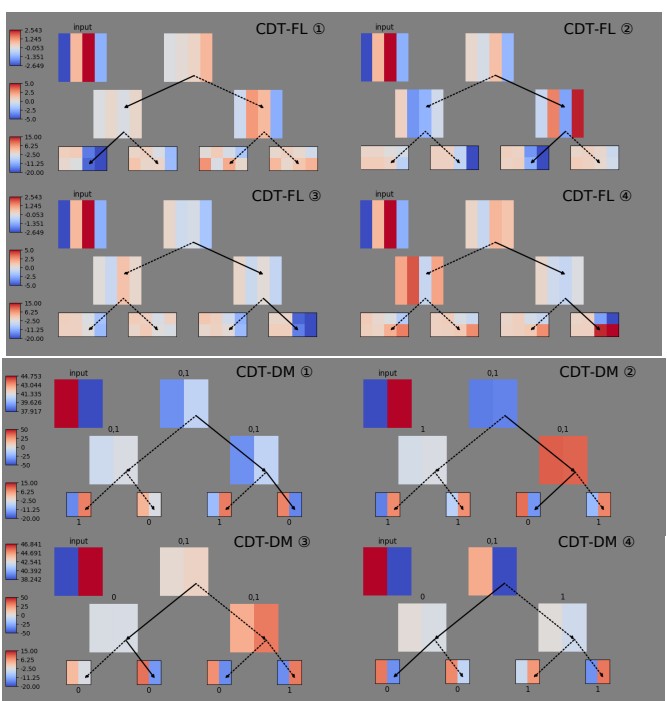

Figure 21: Comparison of four runs with the same setting for CDT (before discretization) imitation learning on *CartPole-v1*: feature learning trees (top) and decision making trees (bottom).

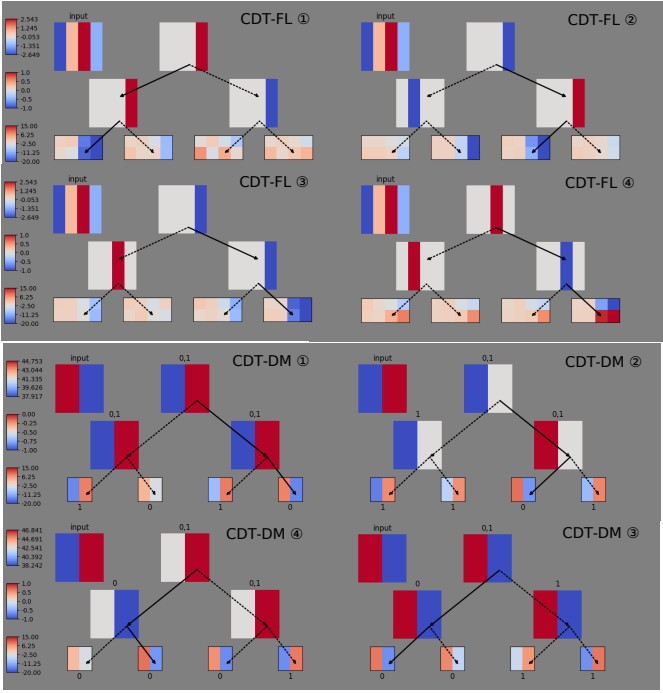

Figure 22: Comparison of four runs with the same setting for CDT (after discretization) imitation learning on *CartPole-v1*: feature learning trees (top) and decision making trees (bottom).

# G    Training Details in Reinforcement Learning

| Tree Type | Env | Hyperparameter | Value |
|---|---|---|---|
| Common | CartPole-v1 | learning rate | $5 \times 10^{-4}$ |
| | | $\gamma$ | 0.98 |
| | | $\lambda$ | 0.95 |
| | | $\epsilon$ | 0.1 |
| | | update iteration | 3 |
| | | hidden dimension (value) | 128 |
| | | episodes | 3000 |
| | | time horizon | 1000 |
| | LunarLander-v2 | learning rate | $5 \times 10^{-4}$ |
| | | $\gamma$ | 0.98 |
| | | $\lambda$ | 0.95 |
| | | $\epsilon$ | 0.1 |
| | | update iteration | 3 |
| | | hidden dimension (value) | 128 |
| | | episodes | 5000 |
| | | time horizon | 1000 |
| | MountainCar-v0 | learning rate | $5 \times 10^{-3}$ |
| | | $\gamma$ | 0.999 |
| | | $\lambda$ | 0.98 |
| | | $\epsilon$ | 0.1 |
| | | update iteration | 10 |
| | | hidden dimension (value) | 32 |
| | | episodes | 5000 |
| | | time horizon | 1000 |
| MLP | CartPole-v1 | hidden dimension (policy) | 128 |
| | LunarLander-v2 | hidden dimension (policy) | 128 |
| | MountainCar-v0 | hidden dimension (policy) | 32 |
| SDT | CartPole-v1 | depth | 3 |
| | LunarLander-v2 | depth | 4 |
| | MountainCar-v0 | depth | 3 |
| CDT | CartPole-v1 | FL depth | 2 |
| | | DM depth | 2 |
| | | # intermediate variables | 2 |
| | LunarLander-v2 | FL depth | 3 |
| | | DM depth | 3 |
| | | # intermediate variables | 2 |
| | MountainCar-v0 | FL depth | 2 |
| | | DM depth | 2 |
| | | # intermediate variables | 1 |

Table 5: RL hyperparameters. The "Common" hyperparameters are shared for both SDT and CDT.

To normalize the states[4], we collect 3000 episodes of samples for each environment with a well-trained policy and calculate its mean and standard deviation. During training, each state input is subtracted by the mean and divided by the standard deviation.

The hyperparameters for RL are provided in Table 5 for MLP, SDT, and CDT on three environments.

---

[4]We found that sometimes the state normalization can affect the learning performances significantly, especially in RL settings.

## H    ADDITIONAL REINFORCEMENT LEARNING RESULTS

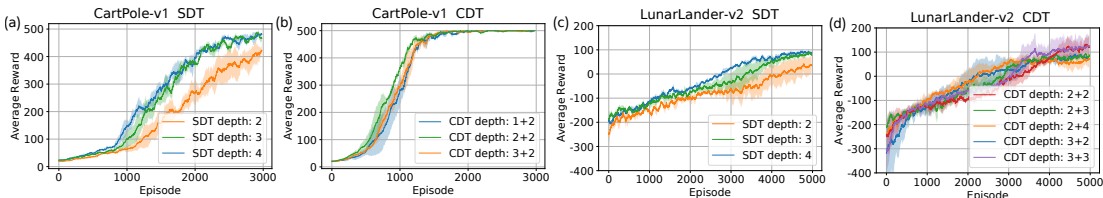

Figure 23: Comparison of SDTs and CDTs with different depths (state unnormalized). (a) and (b) are trained on *CartPole-v1*, while (c) and (d) are on *LunarLander-v2*.

Fig. 23 displays the comparison of learning curves for SDTs and CDTs with different depths, under the RL settings without state normalization. The results are similar as those with state normalization in the main paragraph.

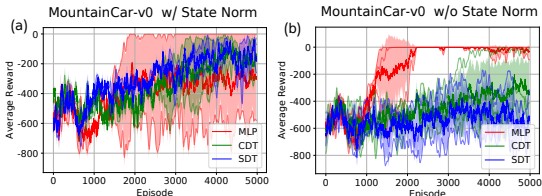

Figure 24: Comparison of SDTs and CDTs on *MountainCar-v0* in terms of average rewards in RL setting: (a) uses normalized input states while (b) uses unnormalized ones.

Fig. 24 shows the comparison of MLP, SDT, and CDT as policy function approximators in RL for the *MountainCar-v0* environment, where the learning curves for each run, as well as their means and standard deviations, are displayed. The MLP model has two layers with 32 hidden units. The depth of SDT is 3. CDT has depths 2 and 2 for the feature learning tree and decision making tree respectively, with the dimension of the intermediate feature as 1. The training results are less stable due to large variances in exploration, but CDTs generally perform better than SDTs with near-optimal agents learned considering both cases.

## I    TREES STRUCTURES COMPARISON

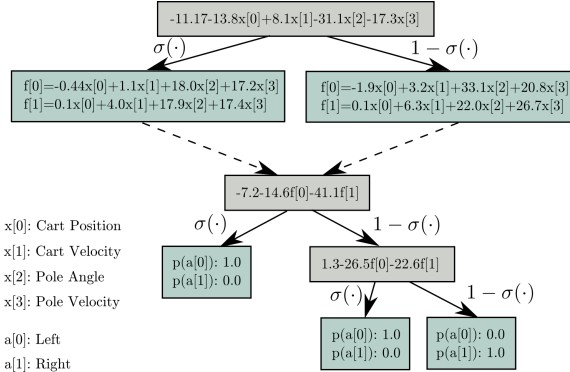

Figure 25: The learned CDT (before discretization) of depth 1+2 for *CartPole-v1*.

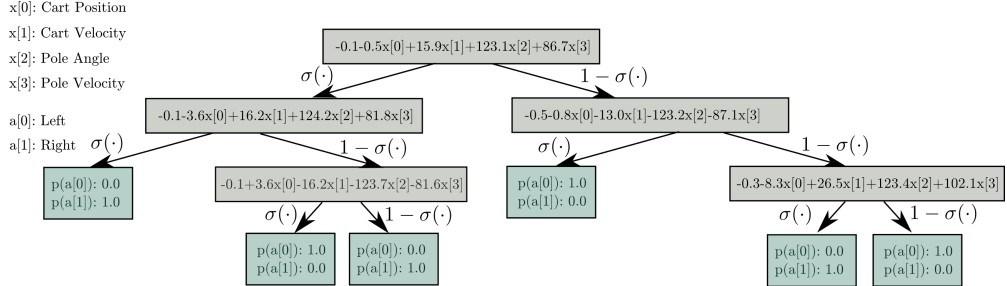

Figure 26: The learned SDT (before discretization) of depth 3 for *CartPole-v1*.

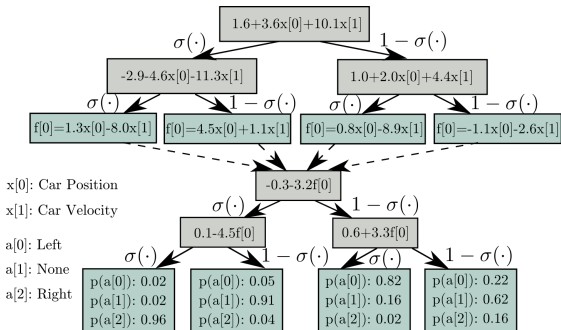

Figure 27: The learned CDT (before discretization) of depth 2+2 for *MountainCar-v0*.

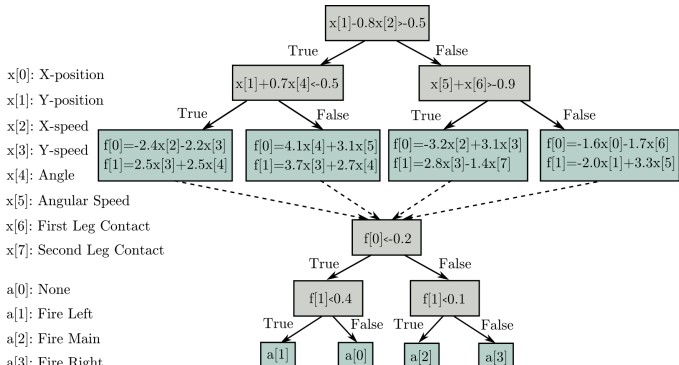

Figure 28: The learned CDT (after discretization) of depth 2+2 for *LunarLander-v2*: two dimensions are reserved for weight vectors in both $\mathcal{F}$ and $\mathcal{D}$, as well as the intermediate features.

