# OpenReview forum: "CDT: Cascading Decision Trees for Explainable Reinforcement Learning"
_ICLR.cc/2021/Conference — Reject_

### Official Review · AnonReviewer4 · 2020-10-28
**CDTs are not demonstrated to be interpretable**

**Rating:** 4
**Confidence:** 5

**Review:**

Edit:

I have read the authors' response and the other reviews. I still believe that this paper is not ready for acceptance.

Summary:

The authors propose to use Cascading Decision Trees (CDTs) to express the policy for an RL agent. The authors describe CDTs and evaluate their use in imitating a trained expert as well as representing a policy during training.


Reasons for score:

The interpretability of CDTs is not convincingly demonstrated by the examples provided. CDTs do perform better than the tested SDTs, but the experiments are insufficient to conclude that CDTs perform well compared to other models which are less interpretable than SDTs.


Pros:

-CDTs are explained well.

-CDTs are shown to solve basic RL environments which are commonly used for evaluation in XRL work.


Cons:

-The authors argue that linear partitions are superior to axis-aligned partitions based on the number of parameters. However, the authors do not consider the number nor complexity of operations required for a "forward pass" through the model. A fairer evaluation of explainability would include the average or worst-case number of operations required to select an action (e.g., 4 multiplications, 4 additions, and 2 comparisons in the worst case for Figure 1a). By using a single D tree for all F trees, the number of parameters is reduced, but the length of any given path is still long (with many operations).

-The authors claim that CDTs are more "explainable" than SDTs, but this is not sufficiently demonstrated. One advantage of DTs over MLPs is that all splitting operations occur on the original features. If the original features are interpretable, then the partitioning process operates on meaningful features. When learned features are used (as in CDTs), this property is lost.

-Between the leaf of a feature learning tree and an internal node in the decision tree, the input is multiplied by a set of weights and put through a non-linearity. When this is performed several times in sequence, this begins to resemble a MLP. The authors do acknowledge this potential problem (as motivation for not evaluating hierarchical CDTs), but this concern also applies (to a lesser degree) to the "single F, single D" case.

-The authors use "heuristic agents" as experts in their experiments. This does not follow the procedure established by prior work, and this is at odds with the motivation of CDTs as useful for explaining RL agents.

-The experimental evaluation is lacking in a number of ways:

--The authors should report policy performance (in imitation learning experiments). Accuracy is a useful metric, but not sufficient on its own. A model can have a high accuracy without learning a well-performing policy. Also, results are reported for a different set of configurations in Table 2 as compared to Table 1 (without any justification). This suggests that discretization of CDTs must be performed in a more nuanced way than otherwise stated in this work.

--The authors do not compare to VIPER in the imitation learning experiments they perform though VIPER was shown to yield higher-performing policies than standard classification-based learning.

--In the RL experiments, the authors compare to a MLP. They find that CDT has a similar (but sometimes worse) final performance despite having fewer parameters. However, the authors should also compare to a smaller MLP, ideally with the same number of parameters as CDT. Without this comparison, conclusions cannot be drawn about the "parameter vs performance" benefits of CDT.

-State normalization (based on a "well-trained policy") is not standard and generally not feasible when applying RL. It is unclear why this was done.

-The second paragraph on page 8 attempts to explain a learned MountainCar-v0 model. The lack of certainty and vagueness of the insights ("kind of like an estimated future position or previous position, and makes action decisions based on that") suggests that additional work is required to make CDTs interpretable. Why are "future position" and "previous position" both options for this explanation? The environment has two features (position and velocity), so discovering that the agent selects actions based on intermediate features derived from position is a given. Ideally, the authors select a way to measure interpretability and perform a quantitative evaluation.


Questions During Rebuttal Period:

Please address and clarify the "Cons" above.


Minor Comments:

-The method's motivation is best placed in the main paper, not in the Appendix (page 3, footnote 2).

-The figures would be more helpful if they appeared on the same page as the corresponding text.

-The caption for Table 1 is not clear with respect to which CDT accuracies are for which discretization schemes.

-The authors note that discretization decreases performance and "claim that this is a general drawback for tree-based methods in XRL..." However, this is only applicable to soft DTs. This should be made clear (e.g., VIPER does not have this drawback).

-The y-limits of Figures 5a and 5b should match so that performances can be more readily compared across plots (given that CDT and SDT are not plotted within one figure). The same applies for Figures 5c and 5d.

-The paper would benefit from another editing pass for grammar.
Some Typos:

-Abstract: "trees (DDTs) have [been] demonstrated to achieve"

-Introduction: "are generally lack[ing] interpretability"; "In this paper, [w]e propose"

---

> ### Author Response · Authors · 2020-11-15
> **Response to Reviewer 4**
>
> We thank the reviewer for the insights and constructive feedback.
>
> Re: Number of operations
>
> It is indeed a good idea to compare the number of different operations in different trees, and we can add analysis of it if necessary. On the other hand, since the nodes in CDT is multivariate, most operations along a path would be multiplications and additions in CDT, which are strongly dependent on the model parameters. In this sense, the comparison of model parameters roughly gives the relationship of number of operations on a typical path. But generally we appreciate the idea.
>
> Re: Explainability of CDT
>
> Our key idea is that the partitioning in original feature space is not enough for generating interpretable model due to the model complexity, as shown in Fig.1. The intermediate feature is necessary although it may lose explainability to some extent, but it will generate some new meaningful representations which are essential for interpret the policy. In this sense, we still find the intermediate features to present some new meanings, for example, the sum of current angle and angular velocity with certain coefficients could represent some expected future angle in CartPole. Therefore the saying that with learned features there will be no meaning is not fair.
>
> Re: Resemble a MLP
>
> It is true that multilayered CDT will resemble the MLP and therefore be lack of interpretability, which is exactly what happens in the ANT method. The key to this problem is to ensure the explainablity of intermediate features in each level of hierarchy. If the explainability of intermediate features is preserved, the multilayer setting will have no problem for CDT. Also, for single $\mathcal{F}$ and single $\mathcal{D}$, the problem is much slighter, because all nodes in $\mathcal{D}$ are just in a space with a single linear transformation from the original input space, regardless of the depth of the tree.
>
> Re: Heuristic agent
>
> We apply the heuristic agent in our experiments for better compare the explainability of CDTs and SDTs against the heuristic agent. We found no other way but to compare the weight vectors in the end. And our experiment tells us that, although the learned tree can mimic the performance of the heuristic agent with such a high accuracy (~$90\%$), the weight vectors are stil quite different from the heuristic agent. The leads the to necessity of XRL with the RL setting rather than imitation learning setting. There are no heuristic agents in RL settings.
>
> Re: Accuracy is not enough
>
> We accept the suggestion to add policy performance beside the accuracy in imitation learning, as shown in the modified paper with means and standard deviations of episode reward added. The difference of Table 1 and Table 2 only lies in that the Table 1 also provides different discretization approaches for CDTs, which involves discretizing the feature learning tree only, the decision making tree only and both of them.
>
> Re: Compare to VIPER
>
> As discussed in response to reviewer 1, the main purpose of the imitation learning experiments is not actually about the high accuracy of CDT both before and after discretization, but the instability of tree-based models for XRL using imitation learning. If the models learned are unstable (not similar for different runs), it doesn't matter how accurate the model can be when considering explainability.
>
> Re: RL experiments
>
> The MLP method in RL experiments is only provided to contectualize the performance of CDT and SDT, without a direct comparison between CDT and MLP. The reason is that even if the number of parameters in MLP is similar as the CDT and SDT, it is still not interpretable due to its fully connected relationship of inner nodes, i.e. not an interpretable model but a black-box model.
>
> Re: State normalization
>
> We doubt the comment that state normalization is not standard in RL. Actually observation normalization is quite common and useful in general RL experiments (see https://arxiv.org/pdf/1709.06560.pdf), sometimes even with significant effects on the learning performances, which is also testified in our experiments. More importantly, since we also evaluate the feature importances for learned trees as in Appendix, the unnormalized raw features will make the feature importance based on the magnitude of weight vectors unfair, which also raises the necessity of investigating the state normalization.
>
> Re: Interpretation of MountainCar
>
> The heuristic solution of MountainCar is actually not quite straightforward to understand, the car needs to have a non-linear function to determine the correct action with its position region and velocity. It's possible that future position or previous position are used for decision.
>
> Re: Minor comments
>
> We appreciate the suggestions and will modify the paper according to that.

---

> > ### Comment · AnonReviewer4 · 2020-11-24
> > **Response to Authors**
> >
> > Thank you for the response to my questions and concerns.
> >
> > Re: Number of operations
> >
> > In general, the number of parameters is not indicative of the number of operations along a path. Additionally, this comment was with respect to the figure (previously labeled Figure 1a) that compared linear partitions with axis-aligned partitions. As noted in the original review, the worst-case is 4 multiplications, 4 additions, and 2 comparisons for the linear boundaries vs 4 comparisons for the axis-aligned tree.
> >
> >
> > Re: Explainability of CDT / Re: Interpretation of MountainCar
> >
> > The difficulty of interpreting the CDT suggests that a CDT is not addressing the desired problem. For MountainCar, "kind of like an estimated future position or previous position, and makes action decisions based on that" is not a valid explanation. When it is unclear whether the agent is using "future" or "previous" position (which is one of two features in that environment), the agent has not been sufficiently explained.
> >
> >
> > Re: Resemble a MLP
> >
> > I believe we are in agreement that CDT lacks interpretability when deep and has diminished interpretability even when shallow.
> >
> >
> > Re: Heuristic agent
> >
> > The heuristic agent is certainly helpful for comparing weight vectors, but an additional experiment with a trained agent would better support the use of CDT.
> >
> >
> > Re: Accuracy is not enough
> >
> > Thank you for adding "Episode Reward" to Table 1!
> >
> >
> > Re: Compare to VIPER
> >
> > A comparison to VIPER should still be performed. I understand the concern about instability in the learned policies, but this needs to be shown for VIPER (along with performance) to demonstrate the trade-off.
> >
> >
> > Re: RL experiments
> >
> > One of the claims made in the paper is that CDTs perform well. This is demonstrated by showing performance for a MLP. In order for the MLP results to provide useful context, appropriately sized MLPs should be used. While noting that CDTs do not learn as fast as MLPs, the authors state that the MLPs have "a significantly larger number of parameters." If the authors would like to use the number of parameters as justification for performance, then they should also include a comparison with a smaller MLP.
> >
> >
> > Re: State normalization
> >
> > 1. It would have been more helpful to point to a specific portion of the 26-page work.
> > 2. This work talks about mean normalization and normalization layers. Neither of these require normalization based on a "well-trained policy", as done for CDT evaluation.
> > 3. My comment was specifically about this "well-trained policy" normalization. It is standard to normalize based on the mean or to obtain features within a desired range.

---

> > > ### Author Response · Authors · 2020-11-25
> > > **Second Response to Reviewer 4**
> > >
> > > Thanks for the detailed response by the reviewer. We would like to address a few points as follows.
> > >
> > > Re: Resemble a MLP
> > >
> > > When saying that "CDT lacks interpretability when deep ...", could the reviewer specify what the 'depth' mean in the context? If the depth of CDT by the reviewer means the numbers of feature-learning and decision-making trees in hierarchical CDT, then it may indeed hurt the interpretability, which is the reason for the lack of interpretability in ANT method. If the depth of CDT means the number of layers in either the feature-learning tree or decision-making tree in a simple CDT (with one feature-learning tree and one decision-making tree), we may not agree with that simple CDT is lack of interpretability compared against normal SDTs.
> > >
> > > Re: Heuristic agent
> > >
> > > The use of CDT is demonstrated to work in the RL setting, where trained agents are compared with CDT. The instability problem happens in the imitation learning setting is generally independent of the agent to mimic from, either heuristic or a trained one.
> > >
> > > Re: State normalization
> > >
> > > Thanks for clarifying the concerns.
> > > The evaluation of the feature importances for learned trees is in Appendix F.
> > > We understand that the normalization based on a well-trained policy is not indeed common to see in RL, while generally (online) normalization based on sampled batches are applied. However, this may lead to varying means and standard deviations of observations in different stages during training. Our goal is to simply normalize each dimension of the observation/feature so that the evaluation of feature importances based on weight vectors can be fair. The normalization should depend on the observation distributions that a well-trained agent will experience, therefore a simple method is to apply an 'offline' normalization based on the observation samples pre-collected by a 'well-trained policy'. A normal way of normalization can be applied with no doubt, but will raise problems like unfair comparison across CDTs trained in different runs.

---

### Official Review · AnonReviewer2 · 2020-10-28
**An appropriate extension of prior differentiable decision tree approaches. Intriguing concepts in nature but claims of improved explainability are not justified and may not be accurate.**

**Rating:** 4
**Confidence:** 4

**Review:**

## After Rebuttal and Discussion Period
I want to first say that I really appreciated the opportunity to review this paper. It was awesome to see the authors willing to respond to the various suggestions and questions that the reviewers provided. The paper has improved over time but some significant areas of improvement remain. There continues to be some discontinuity between the stated scope of this paper as XRL and the proposed CDT approach. During the rebuttal period, I felt that the authors didn't do enough to soften their claims to match the support provided by the contributions present in the paper through both the methods and experimentation. This is not to say that this is not valuable work. As discussed, the concepts and ideas are strong, I however feel that the execution and scoping of this work is a little off from clearly communicating the contributions it makes. One aspect will be through a full evaluation of the utility of their approach as explainable through user studies or other qualitative means. The claims made in this paper regarding explainability are currently unsupported.

Additionally, I believe that there is significant room for improvement in the experimental portion of this work. If there were a way to provide some ablations of the CDT approach as additional baselines as well as the SDT/DDT benchmark, it would significantly improve this portion of the paper.

As it stands, I have not chosen to adjust my score. I do however urge the authors to continue in this line of work. I believe that there is strong merit with the direction they've begun. I look forward to seeing a future completed version of this work.


#### **Summary**
This paper builds from recent developments in differentiable decision tree approaches to address explainable/interpretable Reinforcement Learning. A primary focus of this paper is on the development of rich feature representations to improve the expressivity of the probabilistic splits in the downstream decision nodes of the tree function approximators. Extensive experiments are run to compare the proposed CDT against a general representation of prior differentiable decision tree approaches in two tasks: imitation learning and online policy development.

#### **Assessment**
This paper does a great job introducing the proposed CDT approach in the context of the relevant literature, addressing valid criticisms about prior work. The experiments are extensive, it is clear that a lot of care and thought was put into demonstrating the apparent benefits of CDT. The ideas of cascading improved feature representations to the differentiable decision trees is a reasonable improvement over prior approaches. However, I found the claims of improved explainability to be tenuous at best. Multivariate decision boundaries, and fewer parameters do not generally equate to improved explainability, especially in high dimension input spaces. This challenge of preserving explainability is complicated further by allowing for multiple layers of multivariate decision rules. Even linear functions, when using several variables, lose their ability to be clearly understood after two or three parameters. Traditionally the notion of explainability, especially within RL, is built around the interaction between observed features and the criteria a model uses to arrive at the suggested action. By transforming the decision criteria of the function approximator to be based on a, now uninterpretable, representation of the input, this explainability is no longer preserved. I'd be open to changing my mind if user studies showed the proposed CDT model to be more explainable but the major claims of improved explainability rest on an unsupported conjecture that fewer parameters in linear equations are better. This may have been true in analyzing the results presented in this paper but the experiments are performed over low-dimensional state spaces that have relatively similar oscillatory dynamics admitted by optimal policies. Other weaknesses and suggestions for improvement can be found below.

#### **Strengths**
- The paper does a great job outlining the prior literature in framing the proposed CDT approach. It is clear where the proposed contributions lie within the space of what has been done before. The idea of improving the feature representation for use in a decision tree model is similar to Konstschieder, et al yet the insistence on maintaining model simplicity is refreshing. I'm not convinced that this automatically equates to model explainability but the effort to maintain simplicity is appreciated.
- The paper highlights two central tasks that CDT can be used within RL and extensively evaluates its performance against a generalized form of the prior literature (SDT).
- Within the experimental evaluation the paper evaluates several architectural settings of both CDT and SDT providing a reliable analysis over possible options for the reader to build from.

#### **Weaknesses**
In general, I found the technical development in Section 3 to be unclear. The path $\mathcal{P}$ is described to be a set of nodes yet the only formal definition of $\mathcal{P}$ is in selecting a single node. Within the arg max over possible nodes, it's not clear how a choice of $u$ affects the product of path probabilities. Overall, the discussion about path probabilities is confusing. Continuing, The development for CDT in Section 3.2 only describes a single layer of decision nodes and doesn't account for multiple layers. The transformation matrix $T_{K\times R}$ is only defined for a single decision node. Other points that are unclear throughout section 3 are:
- In the paragraph following Equation 5, it is written: "During the inference process, we simply take the leaf on $\mathcal{F}$ or $\mathcal{D}$ with the largest probability..." Aren't these path probabilities input dependent? Don't they vary stochastically due to the probabilistic definition?
- After equations 6-7, there is some analysis about the reduction of parameters in  CDT. Yet this is hard to place any significance on because a similar analysis is not provided about standard DTs or SDTs. This comparison is also a little tenuous given the different types of SDTs.

Touching on this last point. As defined, SDT is a class of models with very different configurations. It is probably not appropriate to lump them all together in a generalized setting. Even then, the explicit setting of the SDT used to compare with CDT is never described. Without a clear description of the baselines, the experimental results are difficult to fully interpret and place much confidence in. It's not clear why the authors didn't choose several SDT models to evaluate CDT against. In particular, it appears that Silva, et al [AISTATS; 2020] accomplishes a lot of the same goals as CDT albeit without learning a feature representation. It would've been nice to see a more dedicated comparison between CDT and the specific approaches in the literature. More discussion or justification for the use of a single SDT approach is necessary if the current experimental analysis is going to be the final one.

On this point, it is unclear why discretizing the decision trees lead to reduced parameter counts as well as a reduction in overall performance. In most of the prior literature using differentiable or soft decision trees, the discretization is done after training to speed up inference as well as secure the splits at the decision nodes to allow for model explainability. It appears that perhaps the "discretized" form of each of these evaluated models was indicative of how the models were trained? If so, this is a deviation from how the baseline methods were actually developed, leading to an inaccurate comparison. Further clarity about what is mean by discretization here would improve this initial set of experiments greatly.
- Accuracy of an imitation learned model isn't very indicative of how effective that model is in solving the actual task. While informative, the results presented in Tables 1 and 2 don't really communicate how effective CDT or SDT are in solving the tasks. Average return from executing the imitation learned policies would be a better metric here. This would help separate out the differences between a 94% accurate model and a 91% accurate model. Ultimately, we care about how a policy performs on the task. It would be nice to have that analyzed.

- The definition of stability is never formalized. Stability of what? What is the ideal score or metric for stability? Are we looking for the weight vectors to be equivalent between different random initializations of the same tree architecture? If so, I'm not certain that Equation 8 is appropriate as it's measuring the distance a between weight vectors between arbitrary nodes. There is no relation between where in the tree compared weight vectors are coming from. Also, the distance metric will be heavily influenced by the number of parameters and depth of the tree. This makes it an insufficient comparison between tree settings as is done in Table 3.

- In Section 4.2 The results figures over experimental domain are inconsistent. Figures 4 and 5 show results from Cartpole and LunarLander, omitting Mountain Car while Figures 6 and 7 omit LunarLander. Looking at the appendix, it seems that the learning curves for Mountain Car are not as clearly separable (with a lot of variance) and the decision tree for Lunar Lander is not as interpretable. This feels like these results being less aligned with the desired narrative got swept under the rug, hoping that they wouldn't be looked at.

- While the magnitude of weights within the decision node do certainly help determine feature importance, the weights alone doesn't tell me anything about the value of the input and why it leads to one decision being made over another.


#### **Additional Comments**
One paper that I felt was overlooked within the great literature review was:
*Wu, Mike, et al. "Beyond Sparsity: Tree Regularization of Deep Models for Interpretability." AAAI. 2018.*

This omission doesn't negatively affect my estimation of how the authors framed their work but I do feel that it stands to be mentioned alongside other distillation approaches. Not only does this paper distill a Neural Network into a Decision Tree for interpretability, it also regularizes the neural network based on the complexity of that distilled decision tree, ensuring that the end interpretation of the network is simple yet informative.

At present, I do not feel as though this paper is ready for publication. I am not confident in the claims being made by the authors with regards to explainability and would suggest that they either fully justify these claims or place more emphasis on the performance improvements made through CDT at the expense of some explainability. Being more honest with the limitations and assumptions being made in the development of a model is always the best way to go. I do however find the idea of cascading feature representations to encourage more expressive decision nodes within the decision tree to be intriguing. Particularly within the RL space. Representation learning within RL is an open problem and I wonder if this simple modeling strategy may highlight unique aspects of learning representations that deep RL is unable to.

---

> ### Author Response · Authors · 2020-11-15
> **Response to Reviewer 2**
>
> We thank the suggestions from the reviewer and his/her efforts put into this feedback.
>
> Re: Assessment
>
> We admit the comment by the reviewer that multivariate decision boundaries and fewer parameters do not generally equate to improved explainability, which is also discussed in our initial response to the first reviewer. However, from our speculation, explainability generally requires the model compactness as a necessary condition. So what else is missing? Our conjecture is that we also need interpretable intermediate features to form a hierarchical interpretable structure level-by-level, therefore we need to allow our model to construct the intermediate features first. Importantly and surprisingly, the presence of both multivariate decision nodes and intermediate representation learning are missing in most of previous literature based on tree models for explainable models or XRL, which makes the key contribution of our work. Ideally we would like to observe the interpretable intermediate features, while it is found to be handled to some extent by CDT, but not completely. From the stability analysis of the imitation learning models we also found that, the potential model weights as well as intermediate features are not unique to display a certain behavior close to the one representing the training dataset. So further regularization is indeed required to achieve explainable intermediate features or even unique values for weight and features. The reason we haven't yet dug into this direction is that, this regularization to further constraint the model weights may require across-task learning or background knowledge as a prior, considering that a human observing the heuristic solution of LunarLander environment may not be able to infer the proportional controller within it without prior knowledge. This part of idea is left in future work, and CDT is the initial but necessary work to allow that to happen. We know that there are some traditional approaches within XRL using interactions between input features and output actions to interpret the policy, however, we think this is a temporal solution without full potential to interpret a model like a human. This is an okay solution for a given black-box model, but in general we want to interpret/construct the RL policy in the way we write down a heuristic solution. An example to illustrate this is that in the additional paper (Wu, Mike, et al., 2018) mentioned by the reviewer, although the regularization methods can help with aligning the decision boundaries to feature axis and thus make it easy to learn for a univariate decision tree, it could only work for simple classification tasks. Without interpretable intermediate features, it is not possible to fully interpret a complex model with high-dimensional inputs. A typical example for this case is the visual-based task, with images as input for either classification or RL control. The method with only weight regularization will yield an extremely complicated tree structure with boundaries nearly aligned with values of each pixel, which makes no sense for interpretability. The CDT method is an essential cornerstone for handling cases like this.
>
> Re: Definition of $\mathcal{P}$
>
> The formal definition of $\mathcal{P}$ is not for a single node but for a set of nodes denoted as $\{u\}$ rather than $u$ only, which should match with the description the path $\mathcal{P}$ is also a set of nodes. How the choice of $u$ affects the path probabilities lies in $u=2^{i-1}+j$, i.e. through affecting the node indices the path probabilities will change.
>
> Re: Development of CDT in Section 3.2
>
> The mathematical formulation for CDT in Section 3.2 should be sufficient for describing the method. Due to the symmetry in all internal layers within a tree, all internal nodes satisfy the formulas in Eq.(1-3). We do not understand why the reviewer says it only describes a single layer of nodes. Also for the transformation matrix $T_{K\times R}$, since all leaf nodes within the feature learning tree applies an affine transformation for the raw input $x$, this can be summarized into a transformation matrix $T_{K\times R}$ which transforms all leaf nodes in feature learning tree at the same time, rather than only for a single decision node as said by the reviewer.
>
> Re: Input dependent probability in inference
>
> For the sentence "During the inference process, we simply take the leaf on $\mathcal{F}$ or $\mathcal{D}$ with the largest probability..." after Eq.(5), it is true that the path probabilities are input dependent. This sentence describes that during inference, although the paths in tree are stochastic (as the reviewer said) due to the probabilistic definition, we only take the path with largest probability to reach a unique leaf and get a unique action output, rather than averaging over all paths and derive a distribution for the action. This is a similar process as discussed in SDT paper (Frosst and Hinton, 2017).

---

> > ### Author Response · Authors · 2020-11-15
> > **Response to Reviewer 2 (continued)**
> >
> > Re: Reduction of parameters
> >
> > We are sorry to not clearly indicate that the reduction of parameters of CDT is compared against SDT, and we will explicitly phrase it in modification. The reduction of model parameters of CDT is compared against the SDT of the same total depth during the discussion in paragraph, as also detailed in Appendix D.
> >
> > Re: SDT baseline
> >
> > Actually, the SDT baseline in our paper is just the one in Silva et al., 2019 with the same discretization process, where the SDT ideas come from Frosst and Hinton, 2017. We did not find other proper baseline method based on SDT, the VIPER and ANT methods are not proper baselines according to our response to Reviewer 1.
> >
> > Re: Discretization
> >
> > The discretization process strictly follows the description in paper (Silva et al., 2019). The reduction of overall performance can be testified with the models we plotted in Appendix, which are well trained and logged. Since the overall models are simple for some environments like CartPole, it's easy to testify the model with hand-written tree policy give the weights and see how the discretization process hurts the performance.
> >
> > Re: Metric of imitation learning
> >
> > We thank the reviewer for suggesting to provide the average return for different models. This only requires some trivial efforts since we have the models saved. We have already added the episode reward in both Table 2 and Table 4 for further evaluating the imitated models with its rollout performance.
> >
> > Re: Stability
> >
> > The stability is defined to be how similar the models trained with different random initialization and seeds but of the same tree architecture, measured by the average distance of  weight vectors between different instances. The Eq.(8) is indeed measuring the distance of weight vectors between each node in one tree and the closest node in another tree, since we cannot konw which weight vector from a tree would match with which weight vector from another tree beforehand. Generally, this is a reasonable formula for measuring the similarity of decision boundaries between two trees.
> >
> > Re: Result figures
> >
> > We admit that the results in Fig.4,5 and Fig.6,7 are selective, but not for the reason that we do not want them to be looked at. The learning curves for MountainCar are more unstable as pointed out by the reviewer, therefore we put it in Appendix for its lack of representativeness due to variances. However, we can clearly tells that the CDT works better than SDT without state normalization in Fig.24, and not worse than SDT with normalization. The plotting of CDT for LunarLander is put into Appendix simply due to the page limitation.
> >
> > Re: See how decision is made
> >
> > Although the magnitude of weights does not tells the overall decision path, it is sufficient to give the path as long as the input is given. Since there are so many steps in the RL episode, we do not plan to display each step in the paper by giving the input value and displaying decision path.
> >
> > Re: Additional reference
> >
> > Thanks for mentioning the additional paper we missed. We added it in the modified version.

---

### Official Review · AnonReviewer1 · 2020-10-29
**CDT review**

**Rating:** 5
**Confidence:** 2

**Review:**

This paper introduces a new cascading decision trees (CDT) for interpretable RL tasks. As an extension to soft decision trees (SDT), CDT adds a feature learning tree before the decision making tree. By utilizing a low-rank matrix model in the feature learning tree, CDT reduces the parameters of SDT.

1. Novelty is limited. The key novelty of this paper is to add a low-rank representation learning, i.e., feature learning tree, before the decision making tree. Therefore, it is expected that the number of parameters of CDT is less than that of SDT. The improved interpretability is in fact mainly due to such representation learning procedure.

2. It is unclear how to select the tree depths of both the feature learning tree and the decision making tree in a data-driven way. Although the authors provide some preliminary experiments on the tree depth in Figure 5, it is unclear why only $1+2$, $2+2$, $3+2$ are considered in CartPole-v1 and $2+2$, $2+3$, $3+2$, $3+3$ are considered in LunarLander-v2. As shown in Table 5 in Appendix G, the feature learning tree depth, the decision making tree depth, and the number of intermediate variables of CDT are different in three examples.

3. The hierarchical CDT is included in Section 3.2. The authors claimed the hierarchical CDT might be able to improve the prediction accuracy while increasing the model capacity. No experimental study was provided on hierarchical CDT.

---

> ### Author Response · Authors · 2020-11-15
> **Response to Reviewer 1**
>
> We thank the reviewer for reading the paper and her/his feedback.
>
> The contributions of the paper involves leveraging representation learning as pointed out by the reviewer, as well as a proper combination of the feature learning tree and the decision making tree. Additionally, our experiments demonstrate the result that soft-decision-tree-based models with imitation learning for interpreting an RL policy are usually unstable and thus less reliable.
>
> The selection of tree depths and numbers of intermediate variables depends on the complexity of the solution for the environment, whereas the models with proper numbers of parameters are applied.

---

### Official Review · AnonReviewer3 · 2020-10-30

**Rating:** 5
**Confidence:** 3

**Review:**

The paper proposes cascading decision trees (CDTs), which applies representation learning on the decision path of trees, and applies it to explain reinforcement learning models. The main empirical results are that CDTs are claimed to achieve better performance with more compact models than SDTs (soft decision trees), and CDTs as a post-hoc explainer of RL models using imitation learning is unstable.

The results don't seem fully convincing yet. The paper only compares to SDT, although the references mention several other tree-based methods to explain RL models. VIPER would be a nice baseline to add for the imitation learning task, and ANT would also be a nice baseline in general.

Some important results are not yet reported. For example, what is the performance of the black-box model on the imitation learning task? This would contextualize the accuracy numbers of SDT and CDT. Also, discretized SDT has around 50% accuracy while undiscretized SDT has around 94% accuracy, which is a surprising gap. What is the accuracy of a vanilla, hard decision tree on this problem? In Figure 4, CDT doesn't seem much better than SDT for the Lunar Lander problem. Do the authors have any explanation for why this is the case? Perhaps it was mentioned in the paper but I missed it.

Below are some suggested improvements to the presentation of the results:
- For Table 1, perhaps separate out the bracketed results into a different row, e.g. CDT with discretization for the feature learning tree vs. CDTs with only discretization for the decision making tree and discretization for both sub-trees. In fact, maybe 2 columns for SDT — SDT not discretized, SDT discretized — followed by 3 columns for CDT — CDT not discretized, CDT discretization for decision making tree, CDT discretization for both sub-trees. And then make 3 rows: accuracy, depth, # parameters. This is a very detailed suggestion but the gist is that the presentation of this table could be much improved. And similarly for Table 2.
- Why are the Figure 5 cart-pole results distributed over 2 plots? 6 lines on one plot is possible if done with care. Splitting them over 2 plots makes the SDT (Figure 5a) and CDT (Figure 5b) results less comparable, especially when tables are not provided in that section, only figures. The y-axes are also inconsistent, e.g. there is a -300 for the y-axis of Figure 5d but not for 5c, which makes them less comparable.

While the trees being proposed may be more compact, are they more interpretable? I found it hard to parse the linear coefficients and retain all of them in my brain. The trees presented in Figures 6-7 had 4 features, but what if you have more features, or one-hot-encoded features? The literature has user studies to test if human subjects can use trees, but this addition of the linear coefficients to the nodes perhaps warrants a user study to see if human subjects can retain that additional information and make use of it. If a user study is not feasible, could the authors comment on how the interpretability (or lack of) of the feature representation part can be quantified?

I found the findings about instability of trees as a post-hoc explainer interesting and appreciate that the authors including several runs of the methods with the same setting (Figures 15-22). However, I found the figures hard to read, and the heatmap in the tree nodes also needed more explanation, which made me further concerned about the interpretability of the feature representation part of the trees.

Minor points:
- Typo in capitalization in "In this paper, We propose Cascading Decision Trees"
- Citation needed for "some methods have axis-aligned partitions (univariate decision nodes) with much lower model expressivity"
- Some sentences are not precise and perhaps too casual -- “it basically gives a similar solution”, “kind of like an estimated future position”

---

> ### Author Response · Authors · 2020-11-15
> **Response to Reviewer 3**
>
> First of all, thanks for all discussions and suggestions by the reviewer.
>
> As for the problems spotted in the paper, we will provide further explanations as follows, to better illustrate those points which may not be well addressed in paragraphs due to the page limitation, etc.
>
> For the baselines, the method chosen for comparison is thoroughly considered by the authors, to ensure the fairness of comparison. The VIPER and ANT methods are indeed tree-based methods, but they are less fair to be compared with CDT in a specific term of explaining RL policies. The VIPER method learns a non-parametric DTs rather than a parameterized one, and it is only testified in imitation learning settings rather than the reinforcement learning settings. As pointed out in our paper, the experiments in imitation learning demonstrates the instability of tree-based models for XRL, while the reinforcement learning experiments are the actually essential ones to display the explainability of CDTs for RL settings. So comparisons in RL settings would be necessary for VIPER in this sense, which makes it not a proper baseline to display. Apart from this, both the papers of VIPER and ANT do not sufficiently express the explainability of learned models in their contents, especially for ANT, which makes it a model with more efforts put in designing to ensure its accuracy rather than interpretability. With above considerations, the SDT model is the method more suitable to be compared within the topic that CDT is designed for.
>
> We appreciate the reviewer's suggestions in providing more results if necessary. The accuracy drop of SDT models after discretization can be verified in the tree models (with specific values for model weights after training to convergence) we plotted in the appendix. In Fig.4, it's true that the advantages of learning efficiency of CDT over SDT is not significant, our conjecture is that the RL policy in learning is usually task-specific, which may display different behaviours in comparison with different methods, and this is quite common to see in RL literature. However, in general, we indeed observe that the CDT model performs better than SDTs as policy approximators across different environments and settings.
>
> We appreciate the suggested improvements by the reviewer and adopt them in the modified paper. For Fig.5 and Fig.23, we modified the y-axis to keep the two plots for the same environment aligned, but we still separate the SDT and CDT results into two plots because otherwise there will be too many curves on the same plot. The comparison among SDTs and CDTs would be clearer with the aligned axes.
>
> As for the relationship of simplicity of the model (how compact it is) and its interpretability, we have a paragraph of discussion in our draft which is removed in the submitted version. Generally, the simplicity of a model is necessary for its interpretability, but not sufficient. Therefore a more compact model is always preferred, even following the principle of Occam's razor. The hardness of parsing all linear coefficients and keeping them in mind is normal and within our expectation, but this does not proves that the interpretability is not improved in the model. Due to the irreducible complexity of solutions for some tasks (i.e. LunarLander), human can not interpret the policy well even if we provide the heuristic solution to them, as discussed in our motivation example in Appendix B. But the complexity within the duplicative structure of the solution can be reduced by the representation learning in CDT, which is the key to improve interpretability as demonstrated in our method. We indeed find it hard to quantify the interpretability of a model without a user study, which is an open problem in the field.
>
> We add more explanations for the heatmaps of tree nodes in the appendix.
>
> The paper is revised regarding the minor points proposed by the reviewer.

---

### Author Response · Authors · 2020-11-15
**Update Summary**

First we thanks all the reviewers for their insightful suggestions and discussions. We revised the paper in terms of qualitative results in experiments, additional discussion of previous works and clarifying the confusing parts, which are listed as follows:
1. Table 2 and 4 are updated for better demonstrating the experimental results, with episode rewards added for evaluating models and more detailed performance comparison in Table 4 after different discretizations.
2. The axes (y) in Fig.5 and Fig.23 are aligned for better comparison among SDTs and CDTs.
3. An additional paper (Wu et al., AAAI 2018) is added into related work.
4. More explanations of heatmaps/colorbars in the tree plots in Appendix F.3 Fig.15-22 are added.
5. Typos are fixed with an additional editing pass.
6. The placement of figures are improved to keep close with their contexts.
7. Add clarifications for the SDT used in experiments and some mathematical definitions.

---

### Decision · Program_Chairs · 2021-01-07
**Final Decision**

**Decision:**

Reject

**Comment:**

The reviewers and authors have had a significant and healthy discussion around this manuscript. The reviewers remain concerned about the some of the central claims in this manuscript. While they have appreciated the clear communication and willingness of the authors to clarify most of their concerns, this central issue unites the reviewers in maintaining their desire to see a more significant revision of this work before publication. I recommend that the authors take the reviewers' recommendations in improving the presentation and comparison of their ideas.